# Mechano-chemical decomposition of organic friction modifiers with multiple reactive centres induces superlubricity of ta-C

Takuya Kuwahara [1], Pedro A. Romero[1], Stefan Makowski [2], Volker Weihnacht[2], Gianpietro Moras[1] & Michael Moseler [1,3,4]

Superlubricity of tetrahedral amorphous carbon (ta-C) coatings under boundary lubrication with organic friction modifiers is important for industrial applications, but the underlying mechanisms remain elusive. Here, combined experiments and simulations unveil a universal tribochemical mechanism leading to superlubricity of ta-C/ta-C tribopairs. Pin-on-disc sliding experiments show that ultra- and superlow friction with negligible wear can be achieved by lubrication with unsaturated fatty acids or glycerol, but not with saturated fatty acids and hydrocarbons. Atomistic simulations reveal that, due to the simultaneous presence of two reactive centers (carboxylic group and C=C double bond), unsaturated fatty acids can concurrently chemisorb on both ta-C surfaces and bridge the tribogap. Sliding-induced mechanical strain triggers a cascade of molecular fragmentation reactions releasing passivating hydroxyl, keto, epoxy, hydrogen and olefinic groups. Similarly, glycerol's three hydroxyl groups react simultaneously with both ta-C surfaces, causing the molecule's complete mechano-chemical fragmentation and formation of aromatic passivation layers with superlow friction.

[1] Fraunhofer Institute for Mechanics of Materials IWM, MicroTribology Center μTC, Wöhlerstraße 11, 79108 Freiburg, Germany. [2] Fraunhofer Institute for Material and Beam Technology IWS, Winterbergstraße 28, 01277 Dresden, Germany. [3] Physics Department, University of Freiburg, Hermann-Herder-Straße 3, 79104 Freiburg, Germany. [4] Freiburg Materials Research Center, Stefan-Meier-Straße 21, 79104 Freiburg, Germany. Correspondence and requests for materials should be addressed to M.M. (email: michael.moseler@iwm.fraunhofer.de)

Minimisation of frictional energy losses and wear is one basic prerequisite for sustainable mobility[1,2]. For instance, reducing friction in passenger cars to ultralow ($0.1 \geq \mu \geq 0.01$, $\mu$: friction coefficient) and superlow ($\mu \leq 0.01$) levels would significantly lower fuel consumption and global $CO_2$ emission[1]. In this context, the coating of tribological surfaces with diamond-like carbon (DLC)[3] represents an important measure for friction and wear reduction[1]. In particular, hydrogen-free DLC (tetrahedral amorphous carbon, ta-C) has attracted significant attention in the automotive industry[4]. In boundary lubrication experiments, ta-C coatings exhibited superior tribological performance compared to hydrogenated amorphous carbon (a-C:H)[5–7]. A seminal study by De Barros Bouchet and coworkers[5] reported pin-on-disc reciprocating sliding tests of steel/steel, a:C-H/a:C-H, and ta-C/ta-C pairs boundary-lubricated by a mixture of polyalpha-olefin base oil and glycerol-monooleate. While the former two tribosystems exhibited a friction coefficient $\mu > 0.12$, the latter showed ultralow friction $\mu = 0.03$. This study sparked interest into the lubrication of ta-C coatings[6–8] with organic friction modifiers[9]. While boundary lubrication by fatty acids tends to form self-assembled monolayers (SAMs) on a-C:H surfaces[10] and cause ultralow friction ($\mu \approx 0.06$), experiments with an unsaturated fatty acid (oleic acid) between ta-C surfaces yielded ultralow friction ($\mu \approx 0.03–0.07$) under boundary lubrication and even superlow friction ($\mu \approx 0.005$) under mixed lubrication conditions[6,7]. Spectroscopic experiments provided evidence for fragmentation of the oleic acid, passivation of ta-C with hydrogen and hydroxyl groups[6], and additional formation of monolayer-thin aromatic carbon structures[7]. Moreover, Matta et al.[11] observed ultralow friction in reciprocating cylinder-on-disc tests of ta-C boundary lubricated with pure glycerol. Interestingly, a reduction of contact pressure resulted in superlow friction[12]. Accompanying atomistic simulations suggested that H/OH groups originating from glycerol can passivate the ta-C surfaces. These observations indicate that a complete decomposition of the lubricant allows passivation layers to form on top of ta-C, which is a necessary precondition for its superlubricity and nearly wearless sliding[6,7]. Despite these important investigations into the structural details of the lubrication mechanisms, the tribochemical processes underlying the lubricant's fragmentation and the formation of passivating functional groups on ta-C remain elusive. Even less is known about how functional groups and steric hindrance of lubricants promote superlubricity. This lack of understanding makes it impossible to select and design suitable lubricants without numerous heuristic experiments.

In this article, friction and wear experiments combined with extensive quantum and classical molecular dynamics (QMD and CMD) simulations show for the first time that the formation of a superlubricious layer on ta-C coatings under boundary lubrication conditions in the presence of fatty acids and glycerol is caused by a mechanochemical process that relies on the presence of multiple reactive centres in the lubricants.

## Results

A systematic experimental study for four different $C_{18}$ molecules is performed: octadecane, stearic acid, elaidic acid, and oleic acid (see structures in Fig. 1). The lubricants were chosen in order to study the effect of the –COOH carboxylic polar head (by comparing octadecane with fatty acids), the effect of the combined presence of a carboxylic polar head and a –HC=CH– double bond (by comparing stearic acid with elaidic acid and oleic acid) as well as the effect of steric differences between *trans* (elaidic acid) and *cis* (oleic acid) configurations in the allylic rest of the unsaturated fatty acids. For the sequence octadecane → stearic acid → elaidic acid → oleic acid, ta-C shows a clear reproducible ordering in wear (from high wear to negligible wear for elaidic and oleic acid), in friction (from high friction, $\mu = 0.095$, to superlubricity, $\mu = 0.008$) and in $sp^3 \rightarrow sp^2$ rehybridization (measured by the redshift of the Raman G peak ranging from $\Delta f = 10$ to $2\ cm^{-1}$). Since superlubricity was reported also for glycerol-lubricated ta-C[12], we performed additional experiments with glycerol that show that the superlubricious efficiency of oleic acid and glycerol are comparable.

The outcomes of these experiments are rationalized by two kinds of quantum-mechanical atomistic calculations. Quantum molecular statics (QMS) simulations of five different $C_7$ molecules (heptane, heptanoic acid, heptene, *trans*-3-heptenoic acid, and *cis*-3-heptenoic acid) pressurized between ta-C surfaces are performed to study differences in the bonding of these lubricants to ta-C. These simulations show that, at pressure values typical of the experimental conditions, only the unsaturated fatty acids can bridge the tribogap by chemisorption of their two reactive centres (carboxylic group and double-bonded carbon) on the two ta-C surfaces. Conversely, molecules with only one reactive centre (heptanoic acid and heptene) only bind to one of the two surfaces, while saturated heptane molecules exhibit no chemical bonding to the ta-C. Additional QMD sliding simulations reveal that binding of the unsaturated fatty acids to both ta-C counter surfaces results in a transient increase of friction. The related sliding-induced tensile stress on the lubricant molecule causes its rapid step-by-step fragmentation, leading to the release of hydrogen, hydroxyl groups, keto groups, and olefinic fragments, which can passivate the ta-C surfaces and reduce friction and wear.

Large-scale CMD simulations of ta-C asperities sliding on ta-C flat surfaces provide additional insights into the impact of reactive centres on the tribological properties of ta-C. The comparison between octadecane and octadecene lubrication reveals that unsaturated molecules bind more efficiently to the reactive sites on ta-C. This protects the surfaces against cold welding and thus prevents $sp^3 \rightarrow sp^2$ phase transitions and wear.

Finally, additional QMD simulations of glycerol-lubricated ta-C surfaces demonstrate the applicability of these mechano-chemical concepts to other common lubricants and reveal additional insights into the structure of superlubricious tribolayers. The three hydroxyl reactive centres in glycerol promote its complete mechano-chemical decomposition. Interestingly, the resulting chemical intermixing during running-in causes the formation of ultrathin graphene oxide nanostructures. Their aromatic nature allows them to passivate the surfaces without relying on dense chemical terminations. This corroborates earlier experimental results that indicate the importance of aromatic surface passivation in the superlubricity of boundary-lubricated ta-C[7].

**Experiments**. In order to investigate the ability of different $C_{18}$ molecules to passivate and protect ta-C surfaces during tribological shearing, oscillating sliding tests are performed using ta-C-coated pin and disc counter bodies lubricated by octadecane, stearic acid, elaidic acid, and oleic acid (Fig. 1). The experimental conditions were chosen to ensure boundary lubrication for all molecules (see Methods and Supplementary Note 1).

Pronounced differences in pin and disc wear can be observed for the four lubricants. Figure 1a presents micrographs of the final wear tracks on pin and disc after the sliding experiment. Three independent experiments reveal unambiguously that octadecane is a poor lubricant for ta-C: deep wear scars on the disc and even more pronounced on the pin are always observed. The stronger pin wear results from a larger kinematic wear length experienced by the pin. Also lubrication with stearic acid cannot prevent wear

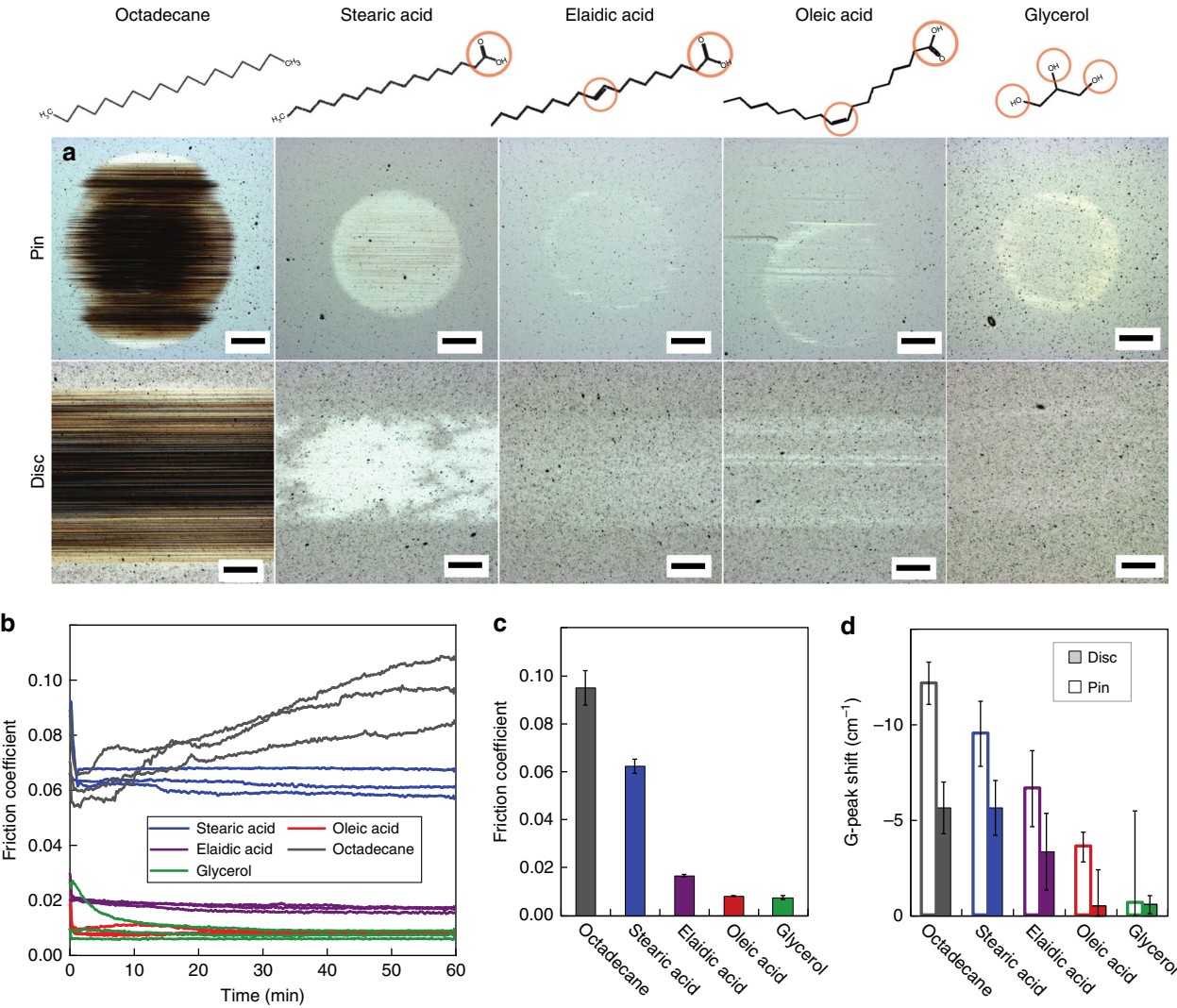

**Fig. 1** Pin-on-disc test: sliding of a ta-C pin on a ta-C disc with octadecane (black), stearic acid (blue), elaidic acid (purple), oleic acid (red), and glycerol (green) lubrication. **a** Micrographs of the wear tracks on pin and disc at the end of a 1-h sliding test (scale bar: 200 μm). **b** Evolution of the friction coefficient for the different lubricants. To ensure the reproducibility, three different tests were performed for each lubricant. **c** Mean friction coefficient $\langle \mu \rangle$ of the last 10 min of the tests. **d** Raman G-peak shift of the ta-C on the pin and disc after sliding compared to before

of ta-C, although wear is strongly reduced compared to the octadecane case. Note, however, that we observed a certain scatter in wear behaviour of stearic-acid-lubricated ta-C. Figure 1a displays one of the two stearic acid tests with noticeable wear. In the third test the wear scar was hardly discernible. Unsaturated fatty acids provide the best surface protection. For elaidic and oleic acid, only a faint impression of the contact area is observable on the disc and the pin. In some cases single wear scars can be observed, in other cases contact area and wear scars are not visible at all. In general, the optical impression of the contact area is not a result of significant material removal but mainly of tribological smoothing of the ta-C surface.

The observed trend in wear protection shows strong correlation with the lubricity of the four $C_{18}$ molecules (see Fig. 1b for the evolution of the friction coefficient $\mu$ and Fig. 1c for the final mean friction $\langle \mu \rangle$ averaged over the last 10 min of the three tests). Octadecane lubrication exhibits a high frictional resistance with $0.08 \leq \langle \mu \rangle \leq 0.11$. After a short running-in and a following short low-friction period with $\mu < 0.07$, that ends after about 5 min of sliding, the friction coefficient starts to rise with superimposed strong fluctuations. Conversely, all fatty acids exhibit a very fast

running-in within less than 50 s followed by a constant friction regime (Fig. 1b). The clear ordering in Fig. 1c, with $\langle \mu \rangle = 0.062$ for stearic acid, $\langle \mu \rangle = 0.017$ for elaidic acid, and $\langle \mu \rangle = 0.008$ for oleic acid, correlates with the wear behaviour in Fig. 1a. There is a pronounced gap in Fig. 1b, c between the saturated molecules (octadecane and stearic acid) with low friction and the unsaturated ones with ultralow (elaidic acid) and superlow friction (oleic acid). Furthermore, the significant difference between elaidic and oleic acid indicates the importance of *trans–cis*-isomerism that results in a different accessibility of the carbon double bond.

Figure 1d reports the Raman G-peak[13] shift of the ta-C after lubricated sliding with the four $C_{18}$ molecules relative to the Raman G-peak of the pristine ta-C surface. The oleic-acid-lubricated ta-C disc and pin exhibit the smallest Raman G-peak shift, suggesting that rubbing the ta-C results in negligible $sp^3 \rightarrow sp^2$ rehybridization within the estimated probing depth of 1 μm. The largest drop in G peak can be observed for stearic acid and octadecane. This suggests a stronger $sp^3 \rightarrow sp^2$ rehybridization for the saturated molecules. The Raman G-peak shift is generally higher on the pin than on the disc, which is attributed to a larger

kinematic sliding length experienced by the pin[6]. We note that, as Raman spectroscopy provides structural information that averages over the depth of the entire coating, the trends observed for the $sp^3 \rightarrow sp^2$ rehybridization do not provide information about a $sp^3 \rightarrow sp^2$ rehybridization in the top carbon layers (graphenisation), whose investigation would require synchrotron based X-ray photoelectron spectroscopy[6,7].

Finally, it is instructive to compare the behaviours of $C_{18}$ molecules and glycerol. The tribological behaviour of glycerol resembles that of oleic acid. Both lubricants exhibit negligible wear on pin and disc as well as a superlow friction of $\mu = 0.008$ after a short running-in phase (Fig. 1a–c). The Raman G-shift tends to be even smaller on the glycerol-lubricated pin (Fig. 1d), while it is almost the same as for oleic acid on the disc.

In summary, the experiments reveal consistent trends for wear, friction, and rehybridization that anticorrelate with the number of reactive sites (–CH=CH–, –COOH, and –OH) on the lubricant molecules. Superlow friction is observed for oleic acid and glycerol. The following three sections report results of atomistic simulations that shed light on the physico-chemical mechanism underlying this ordering and on the role of the lubricant's functional groups.

**QMS and QMD simulations of $C_7$ lubricants**. Quantum-chemical density-functional-based tight-binding (DFTB)[14] simulations are employed to study the chemical reactivity of heptane ($C_7H_{16}$), heptanoic acid ($C_6H_{13}$–COOH), heptene ($C_7H_{14}$) as well as cis-3-heptenoic acid and trans-3-heptenoic acid ($C_6H_{11}$–COOH) confined between two ta-C surfaces (see Lewis structures in Fig. 2a). These molecules have the same chemical functionalities and reactive centres as those used in the experiments but have shorter chain length, which makes computationally intensive quantum-chemical simulations possible. First, QMS simulations are performed with five different ta-C samples. The ta-C/lubricant/ta-C tribosystems are created by cutting a block of ta-C perpendicular to the z-axis, separating the resulting ta-C slabs by 2 nm and introducing a lubricant molecule between the newly created ta-C surfaces. Two thin ta-C layers at the bottom and the top of the system are kept rigid during simulations. Starting from an initial separation $h = 2.15$ nm between the two outer rigid layers, $h$ is gradually reduced, all non-rigid atoms are relaxed and the normal pressure $P_N$ is recorded. Figure 2a shows the resulting pressure-distance curves $P_N(h)$ for the samples 1–5. The colour coding in the $P_N(h)$ curves distinguishes different chemisorption states of the molecule: green represents a free molecule, yellow means chemisorption on one ta-C surface (anchoring), and red indicates chemisorption on both ta-C surfaces (cross-linking). Examples of corresponding interfacial configurations are shown in Fig. 2b (free molecules), Fig. 2c (anchored molecules), and Fig. 2d (cross-linking molecules).

The ta-C surfaces in the five ta-C/lubricant/ta-C tribosystems show considerable variation in their reactivity (defined as the number $N_r$ of reactive carbon atoms in the top surface regions—see Methods section for details). First, we discuss the least reactive ta-C sample (sample 1 with $N_r = 4$ in top row of Fig. 2a). Here, the heptane and heptanoic acid molecules do not react with ta-C surfaces up to high normal pressures ($P_N = 17.7$ and 25.1 GPa, respectively). At higher $P_N$, both molecules release hydrogen atoms and form covalent bonds with both ta-C surfaces. The same happens for heptene, but at a lower pressure $P_N = 5.0$ GPa. Conversely, the trans- and cis-3-heptenoic acid molecules attach to ta-C surfaces via their –COOH polar heads without any external loads. The hydroxyl oxygen releases the hydrogen to a surface dangling bond (DB) and binds to another surface carbon DB via a C–O–C ether bridge (see left panel in Fig. 2c).

Subsequently, at $P_N = 5.2$ GPa (trans) and 3.7 GPa (cis), both carbon atoms involved in the double bond of the unsaturated fatty acid covalently bind to the surfaces and the double bond becomes a single bond (middle panel in Fig. 2d displays a representative structure). Figure 2e, f displays the critical pressures $P_N^c$ for chemisorption on one surface (anchoring) and on two surfaces (cross-linking), respectively. For sample 1 (represented by circles) a clear ordering in these pressures can be seen with high $P_N^c$ for the saturated and low $P_N^c$ for the unsaturated molecules.

The situation is similar for sample 2 (with $N_r = 10$, second row in Fig. 2a), where anchoring and cross-linking can be seen at similar critical pressures (Fig. 2e, f), except for heptanoic acid which shows a strongly reduced critical pressure for anchoring. In this case, the carboxylic head group reacts with a DB site of the surface at $P_N^c = 2$ GPa. Interestingly, the –COOH bonding mechanism is also observed for the more reactive samples 3–5, indicating that a sufficient number of reactive sites must be present to promote this process at low pressures.

A close inspection of Fig. 2a, e, f reveals that the two unsaturated fatty acids (trans- and cis-3-heptenoic acid) show immediate anchoring to all ta-C samples (irrespective of reactivity) and the critical pressures for cross-linking are on average smaller than those for heptane, heptanoic acid, and heptene. These trends can be best seen in Fig. 2g where the sample-averaged critical pressures $\langle P_N^c \rangle$ for anchoring and crosslinking are depicted. On average, anchoring of heptane requires the highest pressures, $\langle P_N^c \rangle = 11.7$ GPa, followed by heptene ($\langle P_N^c \rangle = 2.1$ GPa), heptanoic acid ($\langle P_N^c \rangle = 0.4$ GPa) and both unsaturated fatty acids ($\langle P_N^c \rangle = 0$ GPa). Except for the heptanoic acid case, cross-linking follows the same trend: heptane ($\langle P_N^c \rangle = 16.2$ GPa), heptanoic acid ($\langle P_N^c \rangle = 10.6$ GPa), heptene ($\langle P_N^c \rangle = 6.2$ GPa), and both 3-heptenoic acids ($\langle P_N^c \rangle = 4.0$ GPa).

QMS calculations provide useful basic trends for the chemical reactivity of samples and lubricant molecules. However, the resulting scenario might be incomplete, since tribological loading could trigger further reactions. Therefore, additional sliding QMD simulations of the five lubricant molecules and sample 1 are performed at $P_N = 2$ GPa. Here, the least reactive sample and a relatively small normal pressure are chosen to reflect the mild tribological conditions in the experiments. A detailed chain of chemical reactions for the cis-3-heptenoic acid system is depicted in Fig. 3a (see Supplementary Movie 1 of the QMD trajectory). After cross-linking (hydrogen release from the carboxylic polar head, bonding of the carboxylic radical head to the upper and of a double-bonded carbon to the lower surface at $t = 15$ ps), sliding induces tensile stress in the molecule. The corresponding increase of the frictional force $F_F$ is observed in Fig. 3c. This tensile stretching leads to mechano-chemical breaking[15] of a C–O bond accompanied by the formation of a keto group on the upper surface and of a $C_7H_8O$ radical bonded to the lower surface ($t = 19$ ps). In addition, four hydrogen atoms are transferred to the surfaces. Subsequently, a second cross-linking event occurs via the $C_7H_8O$ radical ($t = 20$ ps). This leads to an additional fragmentation reaction yielding a free CO molecule and a hydroxyl group attached to the upper surface ($t = 22$ ps). The hydroxyl group is the result of a hydrogen transfer from the radical to the surface keto group. The resulting $C_6H_8$ hydrocarbon chain cross-links again at $t = 30$ ps and fragments at $t = 35$ ps causing the formation of a CH group that is incorporated into the upper surface and a $C_5H_7$ unit that is bonded via two C–C bonds to the lower surface. At $t = 58$ ps, the fourth cross-linking event transforms the $C_5H_7$ into an olefinic chain that anchors via a C–C bond to the lower surface. Later on, the free CO molecule binds to the upper surface and forms a keto group. All in all, this trajectory suggests a consumption of the cis-3-

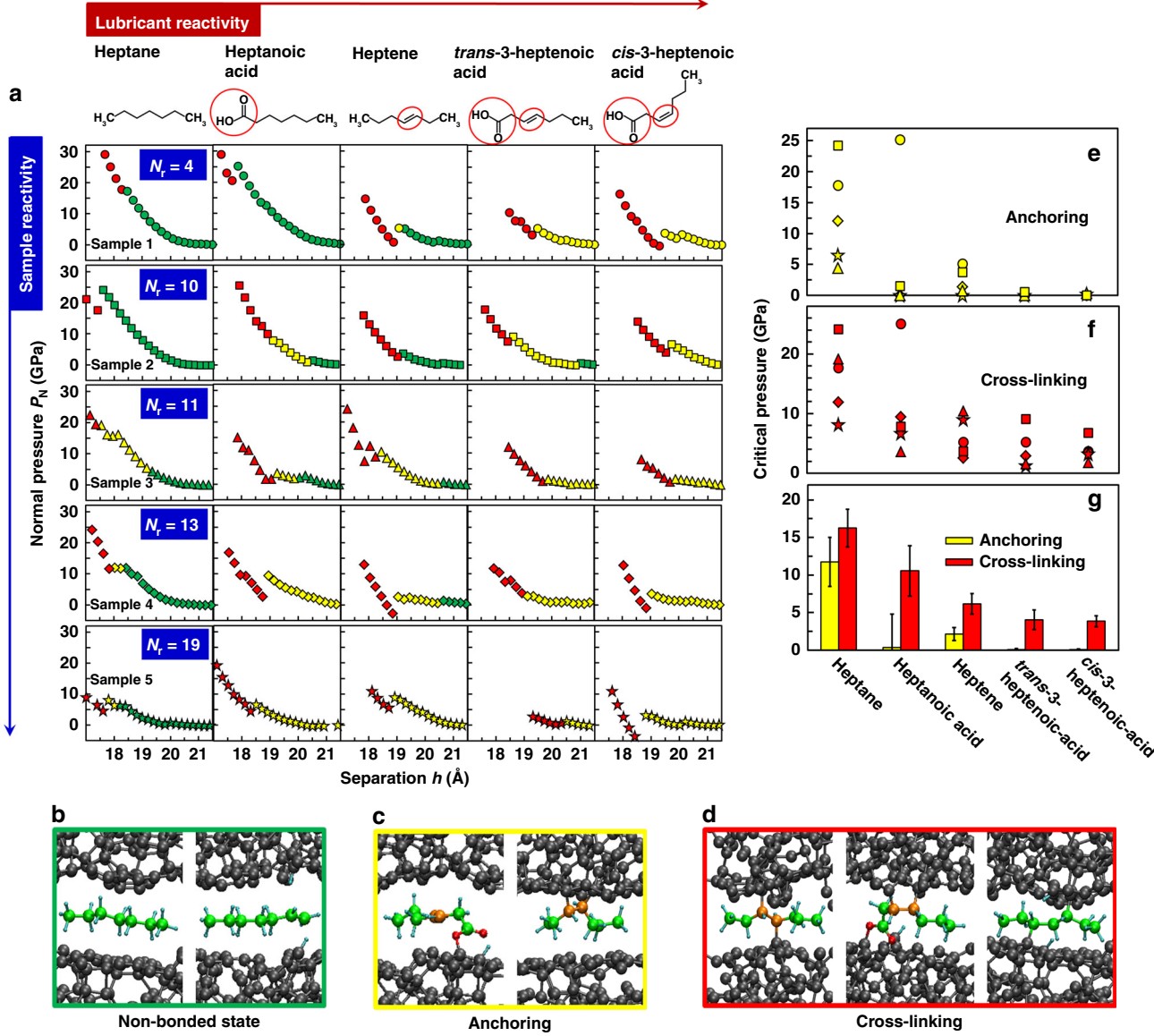

**Fig. 2** Quantum-chemical quasi-static pressing simulations of model lubricant molecules confined between ta-C surfaces. **a** Normal pressure $P_N$ as a function of the separation $h$ between two rigid outer C layers for five ta-C samples. The Lewis structures of the model lubricant molecules are shown in the top row. The order of the five ta-C samples corresponds to the number of reactive surface atoms $N_r$ (see Methods section). Different symbols are employed for five ta-C samples: circle (sample 1), square (sample 2), triangle (sample 3), diamond (sample 4), and star (sample 5). The colour coding distinguishes interfacial phenomena and structures shown in panels **b** and **c**. **b**, **c** Representative snapshots for three types of interfacial chemistry: non-bonded state (green, panel **b**), anchoring (yellow, panel **c**) to one surface, and cross-linking (red, panel **d**) of both surfaces. Grey spheres represent carbon atoms from the ta-C. Green and orange spheres depict initially single- and double-bonded C lubricant atoms, respectively. O and H atoms are in red and cyan, respectively. **e**, **f** Critical pressures $P_N^c$ for the anchoring of lubricant molecules on one ta-C surface (**e**) and for the formation of chemical bonds across two ta-C surfaces (**f**, cross-linking). Panel **g** shows means of the critical pressures in panels **c** and **d**. Error bars represent standard errors of the means. In all panels, the same colour and symbol coding is used

heptanoic acid within a short time. Already after 0.2 ns, the mechano-chemical fragmentation of the molecule produces the following surface terminations: one hydroxyl group, one keto group, three hydrogen atoms, one CH group, and an olefinic $C_5H_7$ (see final configuration in Fig. 3a). Cross-linking and mechano-chemical fragmentation are also observed during a short QMD simulation with oleic acid (Fig. 3b), suggesting the transferability of the $C_7$ result to $C_{18}$ molecules.

Figure 3d displays the final configurations of the other $C_7$ molecules after 0.2-ns sliding. As expected, the two saturated molecules (heptane and heptanoic acid) do not react with the ta-

C at all—in contrast to heptene and *trans*-3-heptenoic acid that anchor to the lower surface already in the early stage of the simulation. Interestingly, both the carboxylic group and the initially double-bonded carbon atoms of *trans*-3-heptenoic acid bind to the lower surface, suggesting that unsaturated fatty acids with *trans*-configuration are less likely to cross-link at low normal pressures.

In summary, the QMS and QMD simulations reveal a pronounced tendency for unsaturated fatty acid to bridge the tribogap by chemically bonding to both ta-C surfaces at moderate normal pressures. In particular, the more pronounced kink in *cis*-

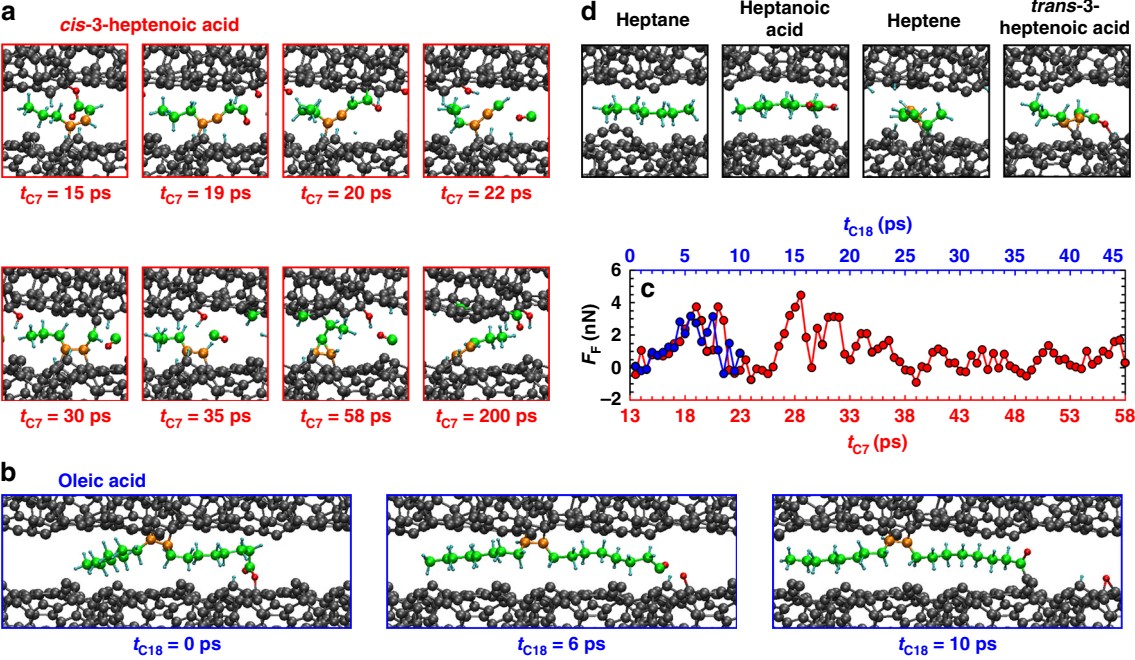

**Fig. 3** QMD simulations of two ta-C surfaces (sample 1) lubricated with various organic molecules. **a** Snapshots (taken at the times $t_{C7}$ indicated below the panels) of the *cis*-3-heptenoic acid trajectory. **b** Snapshots from a 10-ps QMD trajectory of oleic acid at time $t_{C18} = 0$, 6, and 10 ps. **c** Evolution of frictional forces $F_F$ for *cis*-3-heptenoic (red, lower horizontal axis) and oleic acid (blue, upper horizontal axis). The *cis*-3-heptenoic acid (panel **a**) binds to both surfaces around $t_{C7} = 13$ ps, whereas the oleic acid already chemisorbs on both surfaces at $t_{C18} = 0$ ps. To highlight their similar mechanochemistry, $F_F$ is presented from $t_{C7} = 13$ ps for *cis*-3-heptenoic acid on the lower horizontal axis. **d** Final atomic configurations of the other $C_7$ molecules at $t = 0.2$ ns. Green and orange spheres represent the carbon atoms that initially belong to the lubricant molecules. Orange atoms are initially double bonded

3-heptenoic acid represents a steric promoter for cross-linking (see Supplementary Note 2), leading to a fast fragmentation into passivating functional groups. Although the cross-linking is accompanied by high friction during running-in, the related large shear forces induce tensile strain in the molecules (Fig. 3c). This causes their fragmentation and the resulting chemical passivation leads to nearly wearless sliding and superlow friction of ta-C[6].

**CMD simulations**. In this section, we elucidate how unsaturated molecules can protect ta-C surfaces from rehybridization and wear. This question can only be answered on a larger length scale, ruling out the application of QMD. We therefore performed CMD simulations of model tribosystems that represent the contact between rough surfaces (Fig. 4). Lubrication by octadecane and octadecene is considered in order to study two generic model systems representing saturated and unsaturated $C_{18}$ molecules, respectively. In both cases, a sinusoidal model ta-C asperity interacts with a ta-C flat (Fig. 4a, b). The normal load of 5 GPa is mainly carried by the lubricant molecules (green and orange spheres in Fig. 4) that fill the gap between the two ta-C bodies (black spheres in the top two rows of Fig. 4). These simulations show a subtle difference between saturated and unsaturated hydrocarbons (see also Supplementary Movies 2 and 3). In agreement with the QMS and QMD simulations, saturated alkanes show a reduced tendency to anchor to a reactive carbon surface (see close-up in Fig. 4a at 0.29 ns) in contrast to unsaturated lubricants that attach to the ta-C surface via carbon atoms from initially double-bonded sites (orange atoms in the close-up of Fig. 4b). While anchoring protects the ta-C model asperity from wear (compare structures in the bottom row of Fig. 4a, b at 1.0 ns sliding time), octadecane is too mobile and is mostly squeezed out of the contact, failing to protect the two ta-C surfaces from cold welding. In the latter case, covalent bonds between C atoms belonging to the ta-C counter bodies chemically

connect the two surfaces during the whole sliding simulation (cold welding). This can be seen in the bottom row of Fig. 4a, where chemically connected counter bodies are assigned the same (dark grey) colour. The time evolution of the cold welding of the ta-C surfaces is plotted in Fig. 4c (blue curve). This leads to significant adhesive material transfer from the sinusoidal asperity to the flat surface (see wear of asperity in Fig. 4a at 1.0 ns) and to noticeable $sp^3 \rightarrow sp^2$ rehybridization in the surface near ta-C regions (blue curve in Fig. 4d).

Conversely, during sliding, the octadecene molecules anchor via double bonds to the ta-C surfaces, leading to their arrest in the tribogap (close-up in Fig. 4b). The anchoring sites can be easily recognised by looking at octadecene C atoms with three bonds to other C atoms, whose time evolution is plotted in Fig. 4e. As a consequence, both ta-C surfaces are disconnected for longer time periods (see different colours of the counter bodies in Fig. 4b and red curve in Fig. 4c) resulting in a strong reduction of cold welding. This protects the ta-C asperity from wear (see for instance Fig. 4b at 1.0 ns) and mitigates $sp^3 \rightarrow sp^2$ rehybridization (Fig. 4d). Although the CMD simulations lack full chemical complexity, they suggest that more reactive lubricants show a tendency to passivate the surfaces and mitigate wear and rehybridization.

**QMD simulations of glycerol lubrication and aromatic passivation**. Finally, we investigate whether our mechano-chemical mechanism is transferable to other oxygen-containing friction modifiers. For this purpose, we perform a QMD sliding simulation of glycerol-lubricated ta-C at $P_N = 5$ GPa (see Supplementary Movie 4). Initially, a glycerol molecule anchors to the lower surface after releasing one hydrogen atom from a hydroxyl group. After release of the remaining hydroxyl groups, cross-linking occurs (Fig. 5a) and leads to large shear stress $\sigma \sim 1$ GPa for the first sliding phase (Fig. 5b). Subsequently, glycerol completely

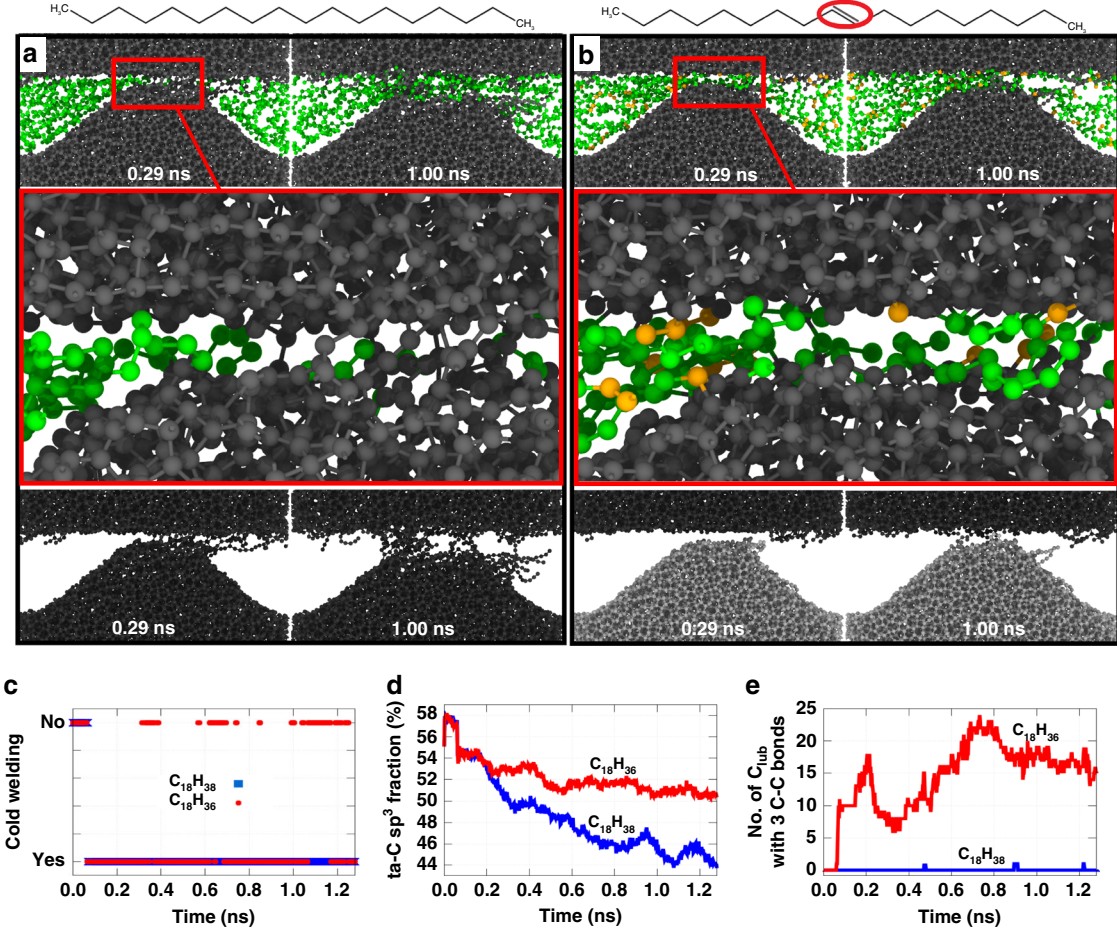

**Fig. 4** Sliding of a ta-C asperity against a flat ta-C surface. The tribocontact is lubricated with **a** octadecane ($C_{18}H_{38}$) and **b** octadecene ($C_{18}H_{36}$) hydrocarbons. In panels **a** and **b**, the first row displays the atoms coloured according to carbon atom type: black spheres represent C atoms in ta-C; green spheres are C atoms in the lubricant; orange spheres are C atoms initially involved in C=C bonds of the octadecene. The second row represents close-up views of the regions marked with the red rectangle in the first row. The third row shows the snapshots from the first row without displaying the lubricant. Here, areas that are not connected by covalent bonds between ta-C atoms have different grey shades. Panel **c** shows the time evolution of cold welding of the two ta-C counter bodies (the chemical connection via ta-C bridges). Panel **d** exhibits the time evolution of the $sp^3$ content in the near-surface ta-C regions. Panel **e** displays the time evolution of the number of lubricant C atoms ($C_{lub}$) that form 3 bonds to other C atoms. These are lubricant C atoms that anchor to the ta-C surface

decomposes and generates passivating species (hydrogen, hydroxyl, and ether groups). Nevertheless, the two surfaces cold-weld after 0.1 ns (while $\sigma$ increases to 10 GPa). Intermixing and rehybridization near the sliding interface promotes formation of dome-shaped aromatic structures (mainly consisting of planar 6-membered rings) that completely hinder chemical bonding across the interface (Fig. 5c), resulting in ultralow friction $\mu = 0.02$ during the last 0.5 ns of sliding. This friction value is higher than the friction measured in the experiments possibly because of the collisions of the dome-shaped nanostructures in Fig. 5c. Indeed, when the upper body is sheared in the perpendicular direction these collisions are avoided and superlow friction ($\mu = 0.008$) is observed in our QMD simulations as well. Thus, it is likely that on experimental time scales extended flat aromatic areas form.

## Discussion

In this study, a combination of complementary experimental and computational approaches shows that a synergy between carbon double bonds and carboxylic polar heads in unsaturated fatty acids promotes the mechano-chemical fragmentation[15] and subsequent formation of superlubricious layers[7] on ta-C. Pin-on-disc sliding tests of ta-C lubricated with octadecane, stearic acid, elaidic acid, and oleic acid yield a reproducible ordering of the friction coefficients and wear volumes (decreasing in this order: octadecane → stearic acid → elaidic acid → oleic acid). Super- and ultra-lubricity of ta-C is achieved in lubrication with the unsaturated fatty acids ($\mu = 0.008$ for oleic acid and $\mu = 0.017$ for elaidic acid), whereas the saturated lubricants yield higher friction ($\mu > 0.06$) and wear.

The experimental observations are rationally explained by quantum-chemical and classical atomistic simulations. QMS simulations predict a diverse interfacial chemistry of the lubricants confined between ta-C surfaces under mechanical loads. The bonding states of the lubricants are classified into three patterns: non-bonded, anchoring and cross-linking. This classification is useful to quantify the reactivity of the lubricants and understand possible scenarios for ta-C lubrication. Octadecane does not react with the surfaces even at high normal pressures ($P_N > 10$ GPa). Stearic acid can anchor via the carboxylic polar head with almost no external loads, but the unreactive, saturated alkyl tail does not allow cross-linking. Thus the adsorption of many stearic acid molecules can lead to the formation of SAMs that prevent cold welding and mitigate friction to some extent.

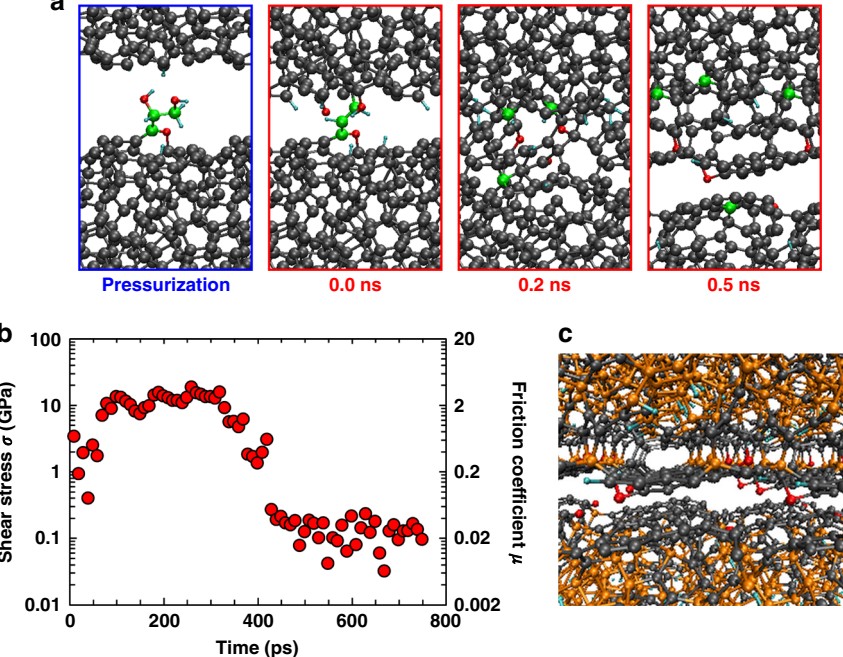

**Fig. 5** A QMD simulation of two ta-C surfaces lubricated with a glycerol molecule. **a** Snapshots of the 1-ns sliding simulation. **b** Evolution of the shear stress $\sigma$ and friction coefficient $\mu$. **c** Shear-induced aromatic passivation of both ta-C surfaces. 3- and 4-fold coordinated C atoms are represented in grey and orange spheres, respectively

The effectiveness of saturated fatty acids for friction reduction is also observed in steel/steel and a-C:H/a-C:H experiments under boundary lubrication[10]. The formation of a close-packed SAM reduces the friction coefficient from 0.15 to 0.10 for steel–steel tribopairs. In a-C:H/a-C:H contacts, regardless of whether saturated fatty acids are present or not, friction coefficient of the order of 0.06 was recorded, in agreement with our results for stearic-acid-lubricated ta-C/ta-C pairs. In the presence of saturated fatty acids, a-C:H surfaces are also covered with a close-packed SAM. Atomistic SAM models predict a higher friction coefficient for elaidic and oleic acid than stearic acid due to their looser packing[16–19]. As a result, this is not a valid model for a super- and ultra-low friction regime of ta-C lubricated with unsaturated fatty acids.

For octadecene, cross-linking is predicted to occur at lower normal pressures compared with the saturated lubricants. Elaidic and oleic acid can cross-link via their carbon double bonds and carboxylic groups at lowest normal pressures ($P_N \approx 4$ GPa). The ordering in the critical pressures $\langle P_N^c \rangle$ for the anchoring and cross-linking is in good agreement with that in the experimentally measured friction coefficient.

QMD sliding simulations demonstrate how the cross-linking via functional groups drives the fragmentation and subsequent formation of superlubricious layers according to the following scenario (Fig. 6). Firstly, an unsaturated fatty acid is likely to anchor on ta-C through the carboxylic polar head (Fig. 6a). High local contact pressures during sliding against a reactive counter body can establish additional anchoring via the double bond reactive centre (Fig. 6b) and cause chemical bonding between the two bodies (cross-linking). Similar tribochemical processes were observed in reactive atomistic simulations of the polymerization of benzene[20] and allyl alcohols[21] and decomposition of phosphorus additives[22]. Yeon et al.[21] suggested that distortions of anchored alcohols accelerate chemical reactions and polymerization occurs at double-bonded carbon and hydroxyl oxygen.

Secondly, shear-induced tensile stretching easily ruptures the cross-linking unsaturated fatty acid. Starting with the release of oxygen atoms due to the preferential rupture of C–O bonds (Fig. 6c), the rapid repetition of the cross-linking and rupture of hydrocarbon chains decreases their length. Simultaneously, the step-by-step consumption and fragmentation of the molecule leads to a dense surface passivation (Fig. 6d) with a variety of functional groups (hydrogen, hydroxyl groups, keto/epoxy groups, olefinic compounds, and presumably aromatic structures). As a result, this process develops a superlubricious monolayer on ta-C. Additional CMD simulations predict the fate of non-anchoring octadecane (saturated). Most of the octadecane molecules can be squeezed out of the contact area, which causes cold welding and subsequent adhesive wear. Furthermore, the cold-welded ta-C junctions experience sizeable elastic and plastic deformation. While the former can give rise to $sp^3 \rightarrow sp^2$ transitions in the bulk of a ta-C film (by inducing rupture of already pre-strained C–C bonds[23]), the latter results in a pronounced $sp^3 \rightarrow sp^2$ rehybridization in the surface-near area[24]. Conversely, octadecene is able to passivate and protect ta-C from complete cold welding. This reduces $sp^3 \rightarrow sp^2$ transitions in the bulk of the coating and restricts the rehybridization to the topmost carbon layers. When comparing this result to our experiments one has to keep in mind that Raman spectroscopy provides average structural information of the entire coating. Since an aromatic passivation of the outermost carbon layers can only be detected by synchrotron XPS[7], our Raman results merely indicate that the superlow friction observed for unsaturated fatty acids and glycerol protects the bulk of the coating from elastic deformations.

These results suggest that the existence of either a carbon double bond or carboxylic group alone in a lubricant molecule is not sufficient to yield superlubricity of ta-C. Their coexistence is a prerequisite for the mechano-chemical fragmentation of the lubricant and formation of superlubricious layers[6,7] on ta-C. Lubricants with one reactive centre (e.g., stearic acid and

**Fig. 6** Schematic representation of the mechano-chemical fragmentation of oleic acid between two sliding ta-C surfaces and of the resulting surface chemical passivation

octadecene) can indeed anchor on ta-C and form SAMs[25]. However, the anchoring of alkenes needs a higher normal pressure. Considering that most of the local contact pressures are well below 1 GPa[6], most of the alkenes should be squeezed out of the tribocontact and a loose-packed SAM cannot completely protect ta-C from cold welding. Although saturated fatty acids can arrange in a close-packed SAM[16,17], relative to ta-C, the softer carpet can give rise to more energy dissipation due to high mobility of hydrocarbon chains[26]. In contrast, mechano-chemical molecular fragmentation during lubrication with unsaturated fatty acids leads to a dense surface passivation with a variety of compact functional groups and reduced energy dissipation. Moreover, although deeper investigations on this aspect are needed, the steric difference between *trans* and *cis* configurations may be explained by our quantum-chemical model. When oleic acid (*cis*) anchors on a ta-C surface via the carboxylic group, carbon double bonds are more exposed to the counter body than in elaidic acid (*trans*) due to its kink structure (see Supplementary Note 2). As a consequence, oleic acid is more likely to be decomposed into fragments through mechano-chemical processes.

The transferability of our mechano-chemical mechanism to other lubricants with multiple reactive centres is demonstrated by experiments and QMD simulations of ta-C boundary-lubricated by glycerol. While the experiments yield the same superlow friction coefficient of $\mu = 0.008$ as for oleic acid, the QMD simulations show that the same mechano-chemical decomposition leads to chemical passivation with hydrogen atoms and oxygen functional groups (hydroxyl and ether). Subsequent cold welding triggers the formation of graphene-like nanostructures at the sliding interface. Their aromatic nature makes them chemically inert and they can withstand high contact pressure without chemical terminations. A typical example of aromatic passivation is a Pandey-reconstructed diamond (111) surface[27,28]. For ta-C, aromatic passivation layers are not perfect graphene, but consist of a mixture of 5-, 6-, and 7-membered rings that are partially oxidized. While material intermixing due to cold welding induces $sp^3 \rightarrow sp^2$ rehybridization (in agreement with dry shearing at diamond/diamond interfaces[29]), it alone cannot form graphitic passivation layers. Hydrogen and oxygen atoms (released from glycerol) terminate carbon DBs, which facilitates the formation of aromatic rings.

In conclusion, this study unveils that the synergy between two functional groups in unsaturated fatty acids promotes the mechano-chemical formation of superlubricious layers on ta-C. Chemical and aromatic surface passivation are responsible for the superlubricity regime, and importantly both require the multiple-reactive-centre-driven mechano-chemical fragmentation. This rule is not limited to $C_{18}$ fatty acids and glycerol but should be universal to any other common lubricants—such as glycerol-monooleate[5] (with two hydroxyl, a carboxyl, and a double bond reactive centre). Therefore, our findings not only enable the prediction of tribological properties of ta-C lubricated with various types of molecules but also provide guidelines for the design of novel organic friction modifiers that drive the formation of superlubricious layers on carbon-based materials.

## Methods

**Experiment**. The lubrication of two ta-C surfaces by $C_{18}$-based hydrocarbons with 0, 1, and 2 reactive centres (–COOH or –HC=CH–) and glycerol is investigated by pin-on-disc sliding tests at low contact pressure (115–160 MPa). Octadecane, stearic acid (octadecanoic acid), elaidic acid (*trans*-9-octadecenoic acid), and oleic acid (*cis*-9-octadecenoic acid) are employed as $C_{18}$ lubricants. Both ta-C surfaces on pin and disc are initially created by coating a low-curvature 100Cr6 spherical pin (about 100 mm radius) and a 100Cr6 disc with a 3.7-μm-thick ta-C film. The Young's modulus (measured by surface acoustic wave spectroscopy[30]) of the films is 430 GPa, corresponding to 53% $sp^3$ hybridization[31] and 2.7 g cm$^{-3}$ density. The deposition is performed with a laser-triggered pulsed arc process[32] (Laser-Arc™). After deposition, the films on both disc and pin are smoothened by diamond slurry lapping to remove hard asperities and growth defects.

Each tribological sliding test is performed for 1 h on a SRV®4-tribometer in reciprocating motion using a stroke length of 5 mm at 10 Hz oscillation frequency and by keeping the normal force $F_N = 50$ N as well as the temperature $T = 80$ °C constant. For these test conditions elasto-hydrodynamic calculations indicate that the experiments for all $C_{18}$ lubricants are performed in the boundary lubrication regime, where the frictional behaviour is governed by ta-C/ta-C or ta-C/lubricant chemical interactions and hardly affected by the lubricant's bulk viscosities (see Supplementary Note 1). Thus, the use of lubricants with the same chain length appeared to us the best strategy to study the impact of chemical functionalities on boundary lubrication behaviour. Conversely, if hydrodynamic lubrication had been of interest, the viscosities of the lubricants should have been matched (e.g., via a chain length adaption).

For each type of lubricant molecule, three different tests with identical conditions are performed to ensure the reproducibility of the measured frictional resistance as well as the ta-C surface degradation. The post-friction surfaces are first inspected by optical microscopy and then characterized by Raman spectroscopy using $\lambda = 514$ nm excitation wave length. Again, measurements from three different areas in the wear scar are taken to ensure reproducibility. The carbon D and G peaks are then fitted to a combined Lorentzian/BWF[13] model within a spectral range of 1000–1800 cm$^{-1}$. The fit quality was very good and resulted in $R^2 > 0.999$. As all measured Raman spectra show no or vanishing D/G-peak ratio ($I_D/I_G < 0.03$), the G-peak position is recorded and its shift relative to the G-peak of the unworn surface is considered as a qualitative measure for $sp^3 \rightarrow sp^2$ rehybridization[33].

**Quantum molecular statics and dynamics**. The chemical reactivity of an alkane, an alkene, an alkanoic acid as well as a *trans* and a *cis* alkenoic acid confined between two ta-C surfaces is studied employing self-consistent-charge DFTB[14,34]. The accuracy of the DFTB Hamiltonian for C–H–O systems has already been demonstrated by some of the authors[27,35]. Five ta-C samples with a density of 2.8 g cm$^{-3}$ are generated by quenching liquid carbon from 8000 to 0 K at a constant rate of $1.0 \times 10^{14}$ K s$^{-1}$. The ta-C/lubricant/ta-C tribosystems are then created by cutting the ta-C samples perpendicular to the z-axis, separating the resulting ta-C slabs by 2 nm and introducing a lubricant molecule into the thus created vacuum region. To simulate surface regions that are chemically reactive due to tribochemical or wear processes, the ta-C surfaces in the tribological contact are not chemically terminated before the simulation. Chemical termination of the tribosurfaces would result in the absence of tribochemical reactions between surfaces and lubricant within the simulation timescale. Conversely, DBs of the outer ta-C surfaces of these tribosystems are terminated with hydrogen atoms to avoid spurious effects and to mimic the presence of bulk ta-C. The hydrogen atoms and neighbouring 0.3-nm-thick ta-C layers are kept rigid during simulations. Three-dimensional periodic boundary conditions are imposed with a system size of $1 \times 1 \times 5$ nm$^3$. Long-range electrostatic interactions are evaluated with the Ewald summation technique[36]. Molecular orbitals (MO) are occupied according to a Fermi–Dirac distribution function with an electronic temperature of 1000 K[37].

Amorphous carbon surfaces exhibit strong inhomogeneities in local atomic configuration and consequently in reactivity. Therefore, an ensemble of ta-C

surface samples is required to obtain stable statistics of the lubricant/ta-C interaction. In this context it is useful to characterise the ta-C surface reactivity by the number $N_r$ of reactive surface sites. To do so, we employ a Mulliken bond order analysis for ta-C surfaces. Within the DFTB theory, the MO $\phi_i$ are expressed as a linear combinations of Slater-type atomic orbitals, $\chi_\mu$: $\phi_i = \sum_\mu c_{\mu i}\chi_\mu$, where $c_{\mu i}$ is the MO coefficient matrix. Integration of the electron density—expressed as the square of the MO at a given point—and summation over all occupied MOs gives the total number of electron in a system,

$$
\begin{aligned}
\sum_i^{\text{OCC}} \int [\phi_i]^2 \, d\boldsymbol{r} \quad &= \sum_i^{\text{OCC}} \int \sum_\mu \sum_\nu \left( c_{\mu i}\chi_\mu \right)\left( c_{\nu i}\chi_\nu \right) d\boldsymbol{r} \\
&= \sum_i^{\text{OCC}} n_i \sum_\mu \sum_\nu c_{\mu i} c_{\nu i} S_{\mu\nu} \\
&= \sum_\mu \sum_\nu D_{\mu\nu} S_{\mu\nu} = N_{\text{elec}},
\end{aligned} \tag{1}
$$

where $D_{\mu\nu}$ and $S_{\mu\nu}$ are the density and overlap matrix and $n_i$ is the occupation number of electrons in the $i$th MO $\phi_i$. Mulliken bond order analysis uses $\mathbf{D} \cdot \mathbf{S}$ for distributing the electrons into atomic contributions. The Mulliken bond order $b_{IJ}$ between atoms $I$ and $J$ is defined by $b_{IJ} = \sum_{\mu \in I} \sum_{\nu \in J} 2D_{\mu\nu} S_{\mu\nu}$.

We consider a carbon atom $I$ as reactive when the sum $M_I = \sum_J b_{IJ}$ of Mulliken bond orders[38] $b_{IJ}$ of its bonds ($IJ$) is lower than a critical threshold $M_I^c$. Here, we choose $M_I^c = 2.16$, i.e., the $M_I$ of a carbon atom on an unsaturated diamond (111) surface. With this definition the five different samples are ordered with increasing reactivity. Sample 1 with $N_r = 4$ is the least reactive, followed by samples 2, 3, and 4 with $N_r = 10$, 11, and 13, respectively. Sample 5 with $N_r = 19$ hosts the highest number of radical sites.

While DFTB accurately describes the interaction of lubricant molecules with ta-C surfaces, its accessible system size is limited to a few hundred atoms. Therefore, quantum-chemical simulations of ta-C surfaces in contact with $C_{18}$ lubricant molecules are not possible due to small periodic cells. Instead of octadecane, stearic acid, oleic acid, and elaidic acid, we model five lubricant molecules with a shorter chain length ($C_7$): heptane ($C_7H_{16}$), heptanoic acid ($C_6H_{13}$–COOH), heptene ($C_7H_{14}$) as well as *cis*-3-heptenoic acid and *trans*-3-heptenoic acid ($C_6H_{11}$–COOH). The corresponding chemical structures are shown in Fig. 2a. These shorter molecules have the same chemical and geometrical characteristics (i.e., –HC=CH– double bonds, –COOH carboxyl groups and *cis-*/*trans*-isomerism) as the $C_{18}$ molecules in the experiments. While effects of the chain lengths on their reactivity cannot be ruled out (e.g., effects due to the different distance between the double bond and the carboxyl group), fundamental insights into the tribochemical mechanisms of fatty acids confined between ta-C surfaces are captured within this DFTB model.

Two types of DFTB simulations are carried out: quasi-static pressing and sliding simulations. In the quasi-static pressing simulations, the upper ta-C slab is moved down in steps of 0.02 nm. At each step, structural optimization is performed until the atomic forces are less than $1 \times 10^{-4}$ a.u. In sliding simulations, the equations of motion are integrated using a velocity Verlet algorithm with time step $\Delta t = 0.5$ fs[36]. The rigid layer of the upper slab is moved horizontally with a constant velocity $v = 100$ m s$^{-1}$ while pressurizing the system with an applied normal pressure $P_N = 2$ GPa using the Pastewka–Moseler barostat[39]. The entire system is thermalized at 300 K using a Langevin thermostat[36] during shearing. The frictional forces $F_F$ on the upper rigid layer at a given time $t$ are averaged during time intervals, providing short-time averages $\langle F_F \rangle$. $\langle F_F \rangle$ is used to determine the frictional shear stress $\sigma = \frac{\langle F_F \rangle}{l_x l_y}$ and friction coefficient $\mu = \frac{\sigma}{P_N}$. Due to limitations of current computational resources, high sliding speeds ($v = 100$ m s$^{-1}$ for QMD and 30 m s$^{-1}$ for CMD) are necessary in order to be able to simulate a long sliding distance and to sufficiently sample phase space. In principle, one must keep the sliding speed well below the speed of sound in materials. The speed of sound in diamond is two or three orders of magnitude larger than our sliding speeds. This ensures that the heat generated in contact during sliding is rapidly dissipated from the sliding interface to surrounding materials. The criterion is a prerequisite for reliable modelling of the shear response of materials and in this sense our simulations are sound. In addition, at low enough temperatures (when thermal activation of reactions can be neglected) the reaction products are entirely determined by the mechano-chemical competition of the mechanical strength of the molecule's different bonds. In this case, mechano-chemical reactions observed in our MD simulations should also be independent of the sliding speed and are thus transferable to our low speed experiments. At finite temperatures, the modeller and the reader have to be aware that reactions are governed by a combination of mechano- and Arrhenius-chemistry. Therefore sliding speed can influence the kinetics of such reactions and it requires some caution when a mechanistic scenario derived from high speed simulations is transferred to low speed experiments.

**Classical molecular dynamics.** While QMD fully resolves the chemical complexity in the lubrication of ta-C with organic friction modifiers, CMD can be used to study chemically less complex lubricants on larger length- and time-scales[40,41]. The lack of reliable interatomic C–H–O force fields dictates the choice of lubricants

that can be studied by the latter approach. Tribological processes in ta-C (such as shear-induced rehybridization[23] or plastic effects in asperities[42]) are realistically described by the second-generation reactive bond order (REBO) potential[43] provided that the interaction range of C–C bonds is truncated by screening functions[44]. Since REBO is restricted to hydrocarbons, fatty acids cannot be studied and only the alkanes and alkenes can be considered in this CMD framework. However, the focus on octadecane and octadecene is sufficient to investigate tribological differences between saturated and unsaturated molecules. A flat ta-C surface sliding against a ta-C sinusoidal asperity is constructed as a simplified model for a local contact interface between rough sliding ta-C surfaces. The region between the two bodies is completely filled with octadecane or octadecene molecules.

This tribosystem is created from ta-C specimen with a volume of $6.65 \times 2.56 \times 9.93$ nm$^3$, a density of 3.10 g cm$^{-3}$ and 62.5% of $sp^3$ carbon atoms. The sinusoidal asperity and the flat surface are carved out of the periodic ta-C specimen by removing the atoms above a sinusoidal surface and below a plane. The resulting system size is $6.65 \times 2.56 \times 13.93$ nm$^3$ and contains 26,272 atoms. The resulting slabs are separated by a 4-nm gap and lubricant molecules (36 octadecane or octadecene molecules) are introduced into the thus created empty space. After rigidly fixing a thin ta-C layer at the bottom and top of the constructed system, the remaining atoms are thermalised to 300 K using a DPD thermostat[45] and a normal pressure $P_N = 5$ GPa is applied to the upper rigid layer[39]. After pressure equilibration, this layer is moved with a constant velocity $v = 30$ m s$^{-1}$. The local pressure at the sinusoidal asperity is almost as large as the applied pressure of 5 GPa. Since the liquid lubricants (octadecane or octadecene) fill the volume around the asperity underneath the upper surface, the applied pressure is well distributed over the asperity and over the fluid lubricant molecules.

**Code availability.** Classical molecular dynamics simulations were performed using the open-source LAMMPS MD package[46]. Quantum-mechanical simulations were conducted using our in-house DFTB code, which is available from the corresponding author upon reasonable request. Visualization was carried out with the open-source softwares OVITO[47] and VMD[48].

## Data availability
The data that support the findings of this study are available from the corresponding author upon reasonable request.

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

## Acknowledgements

The authors gratefully acknowledge funding by the BMWi within the project Pegasus II and the computing time granted by the John von Neumann Institute for Computing (NIC) and provided on the supercomputer JURECA at Jülich Supercomputing Centre (JSC) within projects HFR04 and HFR09. Part of the simulations were performed on the computational resource ForHLR II funded by the Ministry of Science, Research and the Arts Baden-Württemberg and DFG ("Deutsche Forschungsgemeinschaft"). T.K. acknowledges financial support by JSPS Overseas Research Fellowships. S.M. and V.W. thank M. Schwan and F. Schaller for conducting tribological tests.

## Author contributions

M.M. conceived and supervised the research. T.K., P.A.R., G.M., and M.M. designed the simulations. T.K. performed the quantum chemical simulations and P.A.R. performed the classical molecular dynamics simulations. S.M. and V.W. designed and performed the experiments. All authors discussed the results and wrote the manuscript.
