## [Peer Review File · Nature Communications]

Reviewers' comments:

Reviewer #1 (Remarks to the Author):

My opinion is that this paper does not meet the requirements for a publication in this high ranked journal for two major reasons:

- The tribological part is very questionable. The four lubricants chosen for the friction tests have the same hydrocarbon chain length. However they have certainly different viscosity properties so that the lubrication regime in the tests may be significantly different. No fluid film thickness EHL calculations are shown. This kind of comparative tests is often made but with different additive molecules dissolved in the same base oil. It would have been more judicious to adapt the chain length for each species to have the same viscosity at the test temperature in order to highlight specifically the chemical aspect. The main result of the study is the superlow friction obtained with the ta-C tribopair in presence of oleic acid. This result is not new and has already been published in the literature (as referenced by the authors).

- The computer simulation part is certainly interesting and gives additional information on the tribochemical reaction pathways of the shear-induced degradation of the oleic acid molecules. However, the conclusion of this simulation (i.e. the passivation of the carbon atoms by oxygenated terminations) does not give more information than that has already been published from previous experimental investigations. So there is no novelty brought by the paper.

Reviewer #2 (Remarks to the Author):

This manuscript presents experiments and a series of quantum and classical simulations to explore and explain the mechanism by which organic molecules lubricate amorphous carbon surfaces. The overall area is important and interesting, since effective lubrication can have a substantial impact on fuel economy and energy consumption, but there are many unanswered scientific questions on the fundamental mechanisms of friction, wear, and lubrication, especially for systems involving complex chemical reactions. This paper elegantly explains and provides atomistic detail on the highly effective lubricating properties of hydrocarbon molecules, and demonstrates specific criteria for the molecular architecture of such hydrocarbons that is required to endow them with low friction and wear properties. In particular, both a reactive end group and an unsaturated backbone are needed to promote anchoring to one surface followed by trans-interfacial bonding and subsequent scission to form molecular fragments which covalently bond to the hard carbon surface to provide protection from subsequent wear.

While some might argue the conclusions are not surprising, the atomistic detail and the quantification of stress levels required to reach these reactive process thresholds is valuable and novel. I recommend publication. I have only one comment, which is that the authors should more explicitly state the different sliding speeds in the experiments and simulations (0.1 vs 100 m/s) and briefly discuss how this difference (imposed by computational limitations of even the best MD methods) may have some influence on the results. Also, the authors should cite recent work by A. Rappe and J. Yang and coauthors on the stress-induced polymerization of benzene on metal and metal oxide surfaces, which is relevant (and seemingly consistent with) the present work. Finally on page 9, line 200, the authors state "in contrast to heptane" but I believe they mean "in contrast to heptene".

Reviewer #3 (Remarks to the Author):

In this manuscript, combined experiments and simulations have been used to investigate the

tribochemical mechanism leading to superlubricity of ta-C/ta-C contacts lubricated by unsaturated fatty acids.

Pin-on-disc sliding experiments show that super- and ultralow friction with negligible wear can be achieved by lubrication with unsaturated fatty acids, but not with saturated fatty acids and hydrocarbons. Neither the experimental techniques nor the results are particularly novel.

Molecular simulations reveal that, due to the simultaneous presence of a carboxylic group and a C=C double bond, unsaturated fatty acids can concurrently chemisorb on opposing ta-C surfaces. During sliding, the additive remains anchored to both surfaces and the mechanical strain triggers a cascade of molecular fragmentation reactions that release passivating hydroxyl, keto, hydrogen and olefinic groups to the ta-C surfaces. This passivation is proposed as the cause of the superlow friction observed in the macroscopic experiments.

The manuscript is interesting, well written, and the simulation approaches are robust. However, due to computational (QMD) and force-field (CMD) limitations, none of the simulations are performed on the actual molecules of interest in the experiments. Moreover, the accessible timescales are very short and thus it is unclear whether the interesting findings from the simulations are really what is driving the interesting macroscopic behaviour. I remain to be convinced whether passivation alone can explain the superlow macroscopic friction observed, or whether a more complex structural reorganisation (such as that suggested in ref 7) is responsible.

Some more specific comments and recommendations are given below.

1 - Page 2: It should be clarified that Ref 6-8 study OFM friction on both ta-C and DLC surfaces (not just ta-C). It is also important to describe the differences between 'DLC' and 'ta-C' at this stage (see e.g. page 1 of ref 7). The authors also need to be clear that the refs 6-10 are primarily in the boundary lubrication regime, where the load is primarily supported by the asperities, which come into direct contact.

2 - Page 2: Ref 10 studies the friction of alcohols (not fatty acids) - I think that authors meant to cite ref [A] below.

3 - Page 2: The authors describe 0.06 as a 'relatively high friction'; however, this is actually rather low for boundary lubrication conditions. To show how significant the friction reduction is, it would perhaps be useful to place these friction results in the context of traditional steel-steel contacts. I suggest that authors first describe that saturated friction modifiers significantly reduce friction from around 0.15 to 0.10 in steel-steel AFM experiments (ref [A]). In DLC-DLC contacts, the friction coefficient was 0.06 independent of whether saturated friction modifiers were present or not (ref [A]). However, Z-unsaturated fatty acid between ta-C-ta-C contacts showed superlow (0.005) friction (refs 7,8).

4 - Page 2: From their Raman data, the authors suggest that oleic acid 'mitigates sp³→sp² rehybridization'. This assertion seems to be in disagreement with surface analysis in ref 7, which suggested that superlow friction of DLC in presence of oleic acid could be due to 'tribo-formation of a graphene-like structure' (i.e. sp³→sp² at the surface). This important discrepancy needs to be discussed here as well as on page 4. Perhaps oleic acid promotes a small amount of graphitisation at the interface (ref 7) but prevents sp³→sp² transitions in the bulk substrate?

5 - Page 3: In the introduction would be helpful to the reader to mention other molecular simulations of lubricant additive mechanochemistry e.g. refs [B] for 'reactive' CMD and [C] for QMD. It might also be useful to mention other CMD studies of stearic and oleic acid on steel surfaces e.g refs [D,E].

6 - Page 4: The authors declare that unsaturated carboxylic acids give lower friction because they 'can bridge the tribogap by chemisorption of their two reactive centres'. This is counter-intuitive since generally strong chemical bonding to both surfaces prevents sliding and increases friction.

'Bridging' probably means that the mechanical forces on molecules adsorbed to opposing counter-sliding surfaces are greater than if they are adsorbed to one of not adsorbed at all, leading to larger mechanochemical forces and thus faster passivation of the surface (as shown in Figure 3). This needs to be further explained here and in the results section (see e.g. ref [B] which suggests that adsorption is important for mechanochemical reactions to occur).

7 - Page 4: It is noteworthy that the trend shown in Fig 1 is the opposite to that observed for organic friction modifiers in steel-steel contacts (see ref [F]). Specifically, for steel-steel, friction increases from: stearic acid (saturated tail) < elaidic acid (trans-unsaturated) < oleic acid (cis-unsaturated). This is because stearic acid forms close-packed monolayers more readily (see ref [G]) which has been shown in classical MD simulations to give lower friction (see ref [E]). On ta-C surfaces, close-packed monolayer formation is clearly less important to low friction than on steel (ref [A]).

8 - Page 5: In addition to the shorter chain length, the functional groups also have closer proximity to each other in cis-3-heptenoic acid and trans-3-heptenoic acid compared to oleic acid and elaidic acid. This might also be an important factor and should be noted here.

9 - Page 11: The authors suggest that the much lower friction in the cis isomer relative to the trans isomer are because they are 'more exposed to the counter body... due to its kink structure'. From the simulation results, can the authors be any more specific here to rationalise the large macroscopic differences?

10 - Page 14: In addition to GMO and glycerol, could also potentially explain results of ref [H], where superlow friction observed with DLC lubricated by 1,3-Diketone (i.e. two ketone groups)

11 - Page 17: Is the 5 GPa assuming complete contact of the (atomically smooth) slabs? If so, important to indicate that the 'local' pressure at the sinusoidal 'asperities' will be much higher than this.

Suggested Additional References

[A] Simič and Kalin, *Appl. Surf. Sci.*, 2013, 283, 460-470

[B] Yeon et al. *ACS Appl. Mater. Interfaces* 2017, 9, 3142–3148

[C] Loehle et al. *Lubricants* 2018, 6, 31

[D] Doig et al. *Langmuir*, 2014, 30, 186-195

[E] Ewen et al. *Langmuir*, 2016, 32, 4450-4463

[F] Campen et al. *Tribol. Lett.*, 2012, 48, 237-248

[G] Wood et al. *Langmuir*, 2016, 32, 534-540

[H] Amann et al. *ACS Omega*, 2017, 2, 8330-8342

In the following we provide a point-by-point response to the comments of the reviewers. Revisions are marked in blue in this response letter and the revised manuscript.

Reviewer #1:

1) Reviewer's comment:

The tribological part is very questionable. The four lubricants chosen for the friction tests have the same hydrocarbon chain length. However they have certainly different viscosity properties so that the lubrication regime in the tests may be significantly different. No fluid film thickness EHL calculations are shown.

Answer:

We agree with the referee that viscosity is the crucial quantity that should be matched provided the experiments are performed in the hydrodynamic lubrication regime. Since we did not address the lubricant film thickness in our original manuscript, this lack of information led the reviewer to this critical comment as well as the suggestion to match viscosities of the lubricants. In the revised manuscript and the Supplementary Information we present and discuss calculations of elasto-hydrodynamic (EHL) film thicknesses of C₁₈ lubricants in order to determine the actual lubrication regime. We use the Hamrock-Dowson EHL formula to estimate minimum lubricant film thickness h_{\min} for our C₁₈ lubricants as well as for glycerol. These calculations reveal that for all the tribosystems considered in our experiments the values of lambda ($\lambda = h_{\min} / \sqrt{R_{\text{pin}}^2 + R_{\text{disc}}^2}$, where R is the root-mean-square surface roughness of ta-C coated on the spherical pin or disc) are less than unity and, as a result, all the ta-C tribosystems are in the boundary lubrication regimes.

In boundary lubrication the frictional behaviour is hardly affected by the viscosities of lubricants, but is determined predominantly by the ta-C/ta-C and ta-C/lubricant interactions. Our experiments also show that there is no clear correlation between the friction coefficients and lubricant film thicknesses. In particular, the comparison between stearic and oleic acid is the most instructive. The minimum EHL film thickness is significantly larger for stearic acid ($h_{\min} = 12.5$ nm) than for oleic acid ($h_{\min} = 8.2$ nm). Nevertheless, lubrication with stearic acid yields much higher friction and wear than lubrication with oleic acid. This indicates that the frictional behaviour of these systems is governed by the different surface chemistries caused by the different lubricants rather than by their viscosity. As also evidenced by the wear of ta-C and the Raman shifts, asperities are either in direct contact or separated by only a monolayer or a few layers of lubricants, where our atomistic models are valid. A full description of the EHL film thickness calculations was added to Supplementary Information.

“In order to identify actual lubrication regimes under the experimental conditions, minimum film thicknesses of lubricants are calculated using the Hamrock-Dowson formula for elasto-hydrodynamic (EHL) circular contact. Material and experimental parameters is summarized in Table S1. Lubricant viscosities η at 80 °C, pressure-viscosity coefficients α , and calculated EHL film thicknesses h_{\min} are shown in Table 2. The actual lubrication regimes are determined by evaluating the Lambda parameters λ defined as $h_{\min} / \sqrt{R_{\text{pin}}^2 + R_{\text{disc}}^2}$ (boundary lubrication for $\lambda < 1$; mixed lubrication for $1 \leq \lambda < 3$; hydrodynamic lubrication for $3 < \lambda$), where R is the root-mean-square surface roughness of ta-C coated on the spherical pin or disc. The results indicate that all systems are in boundary lubrication, where frictional behaviour is generally not affected by lubricant viscosities. Our experiments also show that there is no clear correlation between the friction coefficients and viscosities. In particular, the comparison between stearic and oleic acid is most instructive. The EHL film thickness for stearic acid is thicker than that for oleic acid, but stearic acid gives much higher friction and wear.”

In addition a hint on the lubrication regime was added to the experimental results section of the revised manuscript.

“For these test conditions elasto-hydrodynamic calculations indicate that the experiments for all C₁₈ lubricants are performed in the boundary lubrication regime, where the frictional behaviour is governed by ta-C/ta-C or ta-C/lubricant chemical interactions and hardly affected by the lubricant’s bulk viscosities (see Supplementary Information for details).”

2) Reviewer’s comment:

This kind of comparative tests is often made but with different additive molecules dissolved in the same base oil. It would have been more judicious to adapt the chain length for each species to have the same viscosity at the test temperature in order to highlight specifically the chemical aspect.

Answer:

Of course, comparing different additives in solution in a base oil, such as polyalpha-olefin (PAO), is an option. However, such an approach would not yield a superlubricity regime for ta-C/ta-C pairs. To the best of our knowledge, superlow friction has been achieved only by lubrication with pure oleic acid or pure glycerol [e.g., De Barros Bouchet et al., *Sci. Rep.* 7, 46394 (2017) and Matta et al., *Phys. Rev. B* 78, 085436 (2008)]. Moreover, a base oil adds additional complexity to the frictional behaviour. For instance, there are reports of tribochemical wear for purely PAO lubricated ta-C coatings [e.g. H.A. Tasdemir et al., *Tribol. Int.* 65, 286 (2013) and C. Héau et al., *Lubr. Sci.* 25, 275 (2013)]. Therefore it would be difficult, if not impossible, to discriminate wear effects caused by the base oil from those caused by the friction modifier. Moreover, it would be hard to rule out tribochemistry between PAO and friction modifier. Thus, an additive solution would make it hard for us to understand the role of chemical functionalities in C₁₈ lubricants.

A matching of viscosities via a chain length adaption is an interesting idea for experiments in the hydrodynamic regime. However, as mentioned above, we ensured that all systems are in boundary lubrication, where the viscosities of lubricants are much less important. Thus, the use of lubricants with the same chain length appeared to us the best strategy to study the impact of chemical functionalities. Therefore, we are sure that our experimental setup in this work is reasonable to achieve our purpose. We added a comment to the revise manuscript.

“Thus, the use of lubricants with the same chain length appeared to us the best strategy to study the impact of chemical functionalities on boundary lubrication behaviour. Conversely, if hydrodynamic lubrication had been of interest, the viscosities of the lubricants should have been matched (e.g. via a chain length adaption).”

3) Reviewer’s comment:

The main result of the study is the superlow friction obtained with the ta-C tribopair in presence of oleic acid. This result is not new and has already been published in the literature (as referenced by the authors).

Answer:

As acknowledged by the referee, we are aware of the previous studies on superlow friction of ta-C lubricated with oleic acid or glycerol [e.g., De Barros Bouchet et al., Sci. Rep. 7, 46394 (2017) and Matta et al., Phys. Rev. B 78, 085436 (2008)]. Our work, however, adds significant new and general aspects to previous works. We provide a general explanation of the mechanism that can trigger superlubricity of ta-C and come up with design rules that can lead to lubricant screening and design. In particular, the novel feature of our work is the systematic comparison of C₁₈ lubricants with a varying number of reactive centres. In this work, we argue that the experiments show a clear reproducible anti-correlation in the friction coefficient (and wear) with the number of reactive centres. This experimental result indicates a significant role of chemical functionalities in organic friction modifiers on the frictional behaviour of ta-C. These systematic experiments motivated us to model mechanochemical processes behind the formation of superlubricious layers providing a convincing mechanistic explanation of the observed ordering. In our opinion we made this already clear in the original manuscript, but since one of the key sentences has changed slightly we cite it here.

“In this article, friction and wear experiments combined with extensive reactive simulations show for the first time that the formation of a superlubricious layer on ta-C coatings under boundary lubrication conditions in presence of fatty acids and glycerol is caused by a mechanochemical process that relies on the presence of multiple reactive centres in the lubricants.”

4) Reviewer’s comment:

The computer simulation part is certainly interesting and gives additional information on the tribochemical reaction pathways of the shear-induced degradation of the oleic acid molecules. However, the conclusion of this simulation (i.e. the passivation of the carbon atoms by oxygenated terminations) does not give more information than that has already been published from previous experimental investigations. So there is no novelty brought by the paper.

Answer:

We appreciate the reviewer’s positive assessment of our atomistic modelling. Of course, it is well known that super- or ultra-low friction can be achieved by surface passivation with hydrogen atoms and oxygen functional groups originating from lubricants. In the revised version of the manuscript, we even show that surface aromatization is another possible way to achieve superlubricity. However, this is not the main finding of this work. Here we focus on why some molecules are more effective than others (and some are not effective at all) in triggering the formation of such passivated, superlubricious layer. In answering this question we unveil the mechanism that leads to the formation of the superlubricious layer as well as the structural/chemical requirements for an effective lubricant. As a result, we are able to provide guidelines that can help recognising and designing molecules with “superlubricant” potential. In particular, our atomistic simulations unveil that “superlubricants” can concurrently adsorb on both ta-C surfaces due to the presence of multiple reactive centres

(i.e., -COOH, -C=C- or -OH). Shear-induced tensile strain triggers a cascade of molecular decomposition reactions, resulting in dense chemical passivation with hydrogen atoms and oxygen functional groups. We presented this result on several conferences and experts in tribo- and mechano-chemistry were excited about the mechanism.

In the revised manuscript, new experiments and simulations for glycerol lubrication not only corroborate our mechano-chemical decomposition mechanism, but also provide further insights into the structure of superlubricious layers. Our QMD simulations of ta-C lubricated with glycerol reveal that ultrathin oxidized graphene-like nanostructures are developed on the top of the ta-C as a result of material intermixing and $sp^3 \rightarrow sp^2$ rehybridization. Their aromatic nature inhibits the formation of chemical bonding across the sliding interface without relying on chemical terminations. Oxygen and hydrogen atoms released from lubricants passivate dangling bonds of carbon atoms underneath the surface, and thus facilitate the formation of sp^2 carbon atoms and planar rings.

We believe that the points mentioned here are novel and robust.

The revised manuscript has been enhanced at various places by additional experiments and simulations with glycerol.

“Since superlubricity was reported also for glycerol-lubricated ta-C¹⁰, we performed additional experiments with glycerol that show that the superlubricious efficiency of oleic acid and glycerol are comparable.”

“Finally, additional QMD simulations of glycerol-lubricated ta-C surfaces demonstrate the applicability of these mechano-chemical concepts to other common lubricants and reveal additional insights into the structure of superlubricious tribolayers. The three hydroxyl reactive centres in glycerol promote its complete mechano-chemical decomposition. Interestingly, the resulting chemical intermixing during running-in causes the formation of ultrathin graphene oxide nanostructures. Their aromatic nature allows them to passivate the surfaces without relying on dense chemical terminations. This corroborates earlier experimental results that indicate the importance of aromatic surface passivation in the superlubricity of boundary-lubricated ta-C⁷.”

“Finally, it is instructive to compare the behaviours of C₁₈ molecules and glycerol. The tribological behaviour of glycerol resembles that of oleic acid. Both lubricants exhibit negligible wear on pin and disc as well as a superlow friction of $\mu=0.008$ after a short running-in phase (Fig. 1a–c). The Raman G-shift tends to be even smaller on the glycerol-lubricated pin (Fig. 1d), while it is almost the same as for oleic acid on the disc.”

“QMD simulations of glycerol lubrication and aromatic passivation. Finally, we investigate whether our mechano-chemical mechanism is transferable to other oxygen-containing friction modifiers. For this purpose, we perform a QMD sliding simulation of glycerol-lubricated ta-C. Initially, a glycerol molecule anchors to the lower surface after releasing one hydrogen atom from a hydroxyl group. After release of the remaining hydroxyl groups, cross-linking occurs (Fig. 5a) and leads to large shear stress $\sigma \sim 1$ GPa for the first sliding phase (Fig. 5b). Subsequently, glycerol completely decomposes and generates passivating species (hydrogen, hydroxyl and ether groups). Nevertheless, the two surfaces cold-weld after 0.1 ns (while σ increases to 10 GPa). Intermixing and rehybridization near the sliding interface promotes formation of dome-shaped aromatic structures (mainly consisting of planar 6-membered rings) that completely hinder chemical bonding across the interface (Fig. 5c) resulting in ultralow friction $\mu = 0.02$ during the last 0.5 ns of sliding. This friction value is higher than the friction measured in the experiments possibly because of the collisions of the dome-shaped nanostructures in Fig. 5c. Indeed, when the upper body is sheared in the perpendicular direction these collisions are avoided and superlow friction ($\mu = 0.008$) is observed in our QMD simulations as well. Thus, it is likely that on experimental time scales extended flat aromatic areas form.”

Reviewer #2:

1) Reviewer’s comment:

This paper elegantly explains and provides atomistic detail on the highly effective lubricating properties of hydrocarbon molecules, and demonstrates specific criteria for the molecular architecture of such hydrocarbons that is required to endow them with low friction and wear properties. In particular, both a reactive end group and an unsaturated backbone are needed to promote anchoring to one surface followed by trans-interfacial bonding and subsequent scission to form molecular fragments which covalently bond to the hard carbon surface to provide protection from subsequent wear.

While some might argue the conclusions are not surprising, the atomistic detail and the quantification of stress levels required to reach these reactive process thresholds are valuable and novel. I recommend publication.

Answer:

We are grateful to the reviewer for acknowledging the importance and the potential impact of our work as well as recommending publication. As detailed below, we addressed the reviewer’s comments and made revisions of the manuscript.

2) Reviewer’s comment:

... the authors should more explicitly state the different sliding speeds in the experiments and simulations (0.1 vs 100 m/s) and briefly discuss how this difference (imposed by

computational limitations of even the best MD methods) may have some influence on the results.

Answer:

As in other atomistic simulations, we employed sliding speeds (100 m/s for QMD and 30 m/s for CMD) that are higher than the experimental one (0.1 m/s). As the referee already anticipated, due to the extremely high computational costs of atomic-scale simulations, the use of such high sliding speed is the only way to be able to simulate sliding distances that are long enough to sufficiently sample the complex phase space. Such accelerated sampling poses two problems.

Firstly, one must keep the sliding speed well below the speed of sound in materials. For example, the speed of sound in ta-C is two or three orders of magnitude larger than our sliding speeds. This ensures that the heat generated in the contact during sliding is efficiently dissipated from the sliding interface to the surrounding material. The criterion is a prerequisite for reliable modelling of the shear response of materials and in this sense our simulations are sound. The energy dissipation is modelled by coupling the dynamics of the atoms with a heat bath.

Secondly, one has to ensure that the mechanisms found for high speed sliding can be transferred to smaller sliding velocities. This is currently the painful question for all QMD and CMD simulations. We believe that the results of our simulations can be extended to applications with a much lower sliding speed. First of all, the anchoring and cross-linking mechanisms found in our QMS simulations are independent of sliding velocity. After cross-linking, the mechano-chemical fragmentation of the lubricant is inevitable and this can be safely concluded irrespective of the sliding speed in the QMD simulations. At low enough temperatures (when thermal activation of reactions can be neglected) the reaction products are entirely determined by the mechano-chemical competition of the mechanical strength of the molecule's different bonds. In this case, the mechano-chemical reactions observed in our MD simulations should also be independent of the sliding speed. However, when reactions are governed by a combination of mechano- and Arrhenius-chemistry, sliding speed can influence the kinetics of such reactions and a mechanistic scenario derived from high speed simulations requires a critical inspection before transferring the mechanisms to low speed experiments. The following brief discussion about the discrepancy of the sliding speeds between our experiments and simulations were added to the method section of the revised manuscript.

“Due to limitations of current computational resources, high sliding speeds ($v = 100$ m/s for QMD and 30 m/s for CMD) are necessary in order to be able to simulate a long sliding distance and to sufficiently sample phase space. In principle, one must keep the sliding speed well below the speed of sound in materials. The speed of sound in diamond is two or three orders of magnitude larger than our sliding speeds. This ensures that the heat generated in contact during sliding is rapidly dissipated from the sliding interface to surrounding materials. The criterion is a prerequisite for reliable modeling of the shear response of materials and in

this sense our simulations are sound. In addition, at low enough temperatures (when thermal activation of reactions can be neglected) the reaction products are entirely determined by the mechano-chemical competition of the mechanical strength of the molecule's different bonds. In this case, mechano-chemical reactions observed in our MD simulations should also be independent of the sliding speed and are thus transferable to our low speed experiments. At finite temperatures, the modeller and the reader have to be aware that reactions are governed by a combination of mechano- and Arrhenius-chemistry. Therefore sliding speed can influence the kinetics of such reactions and it requires some caution when a mechanistic scenario derived from high speed simulations is transferred to low speed experiments.”

3) Reviewer's comment:

... the authors should cite recent work by A. Rappe and J. Yang and coauthors on the stress-induced polymerization of benzene on metal and metal oxide surfaces, which is relevant (and seemingly consistent with) the present work.

Answer:

Many thanks for bringing this interesting article to our attention. The authors studied mechanochemistry of benzene between two metal surfaces. Their ab-initio simulations revealed that a high normal pressure is necessary for the decomposition of benzene molecules. At high pressures (> 20 GPa), due to the loss of a hydrogen atoms, carbon atoms in the 6-membered rings become reactive and polymerize. In our simulations, the loss of hydrogen atoms in saturated molecules also creates reactive sites and facilitates anchoring and cross-linking at higher pressures. We added the following sentence to relate our findings to previous computational studies of mechanochemistry of organic molecules to the discussion of the revised manuscript.

“Similar tribochemical processes were observed in reactive atomistic simulations of the polymerization of benzene¹⁸ and allyl alcohols¹⁹ and decomposition of phosphorus additives²⁰. Yeon et al.¹⁹ suggested that distortions of anchored alcohols accelerate chemical reactions and polymerization occurs at double-bonded carbon and hydroxyl oxygen.”

4) Reviewer's comment:

Finally on page 9, line 200, the authors state "in contrast to heptane" but I believe they mean "in contrast to heptene".

Answer:

We thank the referee for pointing out this typo. We corrected it in the revised manuscript.

Reviewer #3:

1) Reviewer's comment:

The manuscript is interesting, well written, and the simulation approaches are robust. However, due to computational (QMD) and force-field (CMD) limitations, none of the simulations are performed on the actual molecules of interest in the experiments. Moreover, the accessible timescales are very short and thus it is unclear whether the interesting findings from the simulations are really what is driving the interesting macroscopic behaviour. I remain to be convinced whether passivation alone can explain the superlow macroscopic friction observed, or whether a more complex structural reorganisation (such as that suggested in ref 7) is responsible.

Answer:

We thank the reviewer for the positive assessment of our manuscript. Due to the extremely high computational cost of QMD simulations, it is a challenge to generate reliable statistics with C₁₈ lubricants. However, we agree that it is important to verify the transferability of the tribochemical mechanism obtained from simulations of C₇ model lubricants to C₁₈ lubricants. Therefore, we performed additional QMS and QMD simulations of C₁₈ fatty acids (stearic, elaidic and oleic acid). Although only one QMS trajectory for each molecule was generated, the results are in agreement with the trends observed for C₇ molecules. Oleic acid can chemisorb on both ta-C surfaces at low normal pressure of 2.84 GPa. The critical pressure for cross-linking is close to that in cis-3-heptenoic acid. A QMD simulation of oleic acid also shows the same mechano-chemical fragmentation facilitated by the presence of two reactive centres (–COOH polar head and C=C double bond). We added the results obtained for C₁₈ fatty acids to Fig. 3 and Supplementary Information.

The comment on the atomic-scale mechanism behind the macroscopic superlow friction turned out to be very useful. While chemical passivation is supported by other experimental and theoretical studies, structural reorganization (graphenisation of the top surface layers) is also observed by state-of-the-art measurements (e.g. Ref. [7]). Thus, actual structures of superlubricious layers supported by ta-C remain controversial.

Since we did not observe full decomposition of unsaturated fatty acids within the time scale of our QMD simulations, we decided to enhance our lubricant test set by adding a much shorter molecule. We chose glycerol and fortunately the new experiments in the revised manuscript revealed that glycerol has a frictional behaviour that is very similar to oleic acid (i.e. our new sliding tests of ta-C lubricated with glycerol yielded superlow friction with $\mu = 0.008$). At the same time we carried out 1-ns QMD simulations of glycerol-lubricated ta-C. During these longer QMD simulations a full mechano-chemical decomposition of glycerol could be observed resulting in the formation of ultrathin oxidized graphene-like tribolayers on both ta-C tribo partners. Initially, the mechano-chemical decomposition is triggered by the same process (anchoring \rightarrow cross-linking) as for unsaturated fatty acids. After complete decomposition of glycerol cold-welding occurs, leading to material transfer and chemical intermixing. Interestingly, mechanical rubbing breaks C–C bonds, and divalent oxygen and monovalent hydrogen atoms terminate carbon dangling bonds, which facilitates the formation of *sp*² carbon atoms, planar aromatic rings, and planar graphene-like layers near the sliding interface. As a result, oxidised graphene-like nanostructures form on both surfaces, accompanied by a two order of magnitude reduction in shear stress (towards a superlubricity level). The aromatic nature of the graphene-like tribolayers enables them to

support a high mechanical load without relying on dense chemical passivation. We suspect that aromatic passivation is not unique to glycerol lubrication, although we were able to observe this in the glycerol case due to its limited size and high oxygen content. In general, we believe that the complete decomposition of lubricant molecules and large mechanical forces during material intermixing are prerequisites for its occurrence. The result of glycerol-lubricated ta-C is only a representative example of how graphene-like tribolayers form and give an idea about their structure. Due to limitations of time and spatial scale, no material intermixing and aromatic surface phase transitions are observed for cis-3-heptenoic acid.

Finally, the glycerol result indicates that both dense chemical passivation and aromatic passivation can induce ultra- and even superlow friction of ta-C. Importantly, both require a cascade of mechano-chemical decomposition reactions. At least two reactive centres in lubricants are necessary to concurrently chemisorb on both surfaces and thus facilitate the decomposition, which is a main finding of this study. On this point, since three hydroxyl groups in glycerol can be considered as multiple reactive centres, we conclude that the multiple-reactive-centre-driven mechano-chemistry is applicable to other common lubricants. The simulation result was added to Fig. 5 of the revised manuscript and the new findings are discussed at various places in the revised manuscript. The reviewer is referred to our response to comment 4) of reviewer #1 for an explicit list of the changes.

2) Reviewer's comment:

Page 2: It should be clarified that Ref 6-8 study OFM friction on both ta-C and DLC surfaces (not just ta-C). It is also important to describe the differences between 'DLC' and 'ta-C' at this stage (see e.g. page 1 of ref 7).

Answer:

This is a helpful comment. As the reviewer noted, Refs. [7,8] studied tribological properties of both ta-C/ta-C and a-C:H/a-C:H pairs in oleic acid lubrication. We changed the introduction in order to clearly distinguish between hydrogen-free and hydrogenated DLCs.

“In this context, the coating of tribological surfaces with diamond-like carbon (DLC)³ represents an important measure for friction and wear reduction¹. In particular, hydrogen-free DLC (tetrahedral amorphous carbon, ta-C) has attracted significant attention in the automotive industry. In boundary lubrication experiments, ta-C coatings exhibited superior tribological performance compared to hydrogenated amorphous carbon (a-C:H)^{4,5,6}. A seminal study by De Barros Bouchet and coworkers⁴ reported pin-on-disc SRV experiments of steel/steel, a-C:H/a-C:H and ta-C/ta-C pairs boundary-lubricated by a mixture of polyalpha-olefin base oil and glycerol-monooleate. While the former two tribosystems exhibited a friction coefficient $\mu > 0.12$, the latter showed ultralow friction $\mu = 0.03$.”

3) Reviewer's comment:

The authors also need to be clear that the refs 6-10 are primarily in the boundary lubrication regime, where the load is primarily supported by the asperities, which come into direct contact.

Answer:

We absolutely agree that we should mention that Refs. [6-10] studied tribological properties of ta-C in boundary lubrication. The EHL film thickness calculations of C₁₈ lubricants that we added to the revised manuscript show that also our systems are in boundary lubrication, where the friction behaviour is not affected by the bulk viscosities of lubricants but determined predominantly by the ta-C/ta-C or ta-C/lubricants interactions. Our experiments also show that there is no correlation between the friction coefficients and the viscosities of lubricants. As shown in this study and previously in Refs. [6-10], tribo-induced structural/chemical modifications of ta-C surfaces significantly affect the friction behaviour. Whenever appropriate we added the expression “boundary lubrication” or “boundary-lubricated” to the introduction of the revised manuscript.

4) Reviewer’s comment:

Page 2: Ref 10 studies the friction of alcohols (not fatty acids) - I think that authors meant to cite ref [A] below.

Answer:

We thank the reviewer for noting the citation error. We corrected it in the revised manuscript.

5) Reviewer’s comment:

Page 2: The authors describe 0.06 as a 'relatively high friction'; however, this is actually rather low for boundary lubrication conditions. To show how significant the friction reduction is, it would perhaps be useful to place these friction results in the context of traditional steel-steel contacts. I suggest that authors first describe that saturated friction modifiers significantly reduce friction from around 0.15 to 0.10 in steel-steel AFM experiments (ref [A]). In DLC-DLC contacts, the friction coefficient was 0.06 independent of whether saturated friction modifiers were present or not (ref [A]). However, Z-unsaturated fatty acid between ta-C-ta-C contacts showed superlow (0.005) friction (refs 7,8).

Answer:

This is an excellent idea. An effect of saturated fatty acids on the friction properties of other tribosystems (steel/steel and a-C:H/a-C:H) is worth a discussion. For steel/steel pairs, the friction reduction from 0.15 to 0.10 is explained by the formation of a close-packed self-assembled monolayer (SAM). In contrast, for a-C:H/a-C:H pairs, regardless of whether saturated fatty acids are present or not, the same friction coefficient of 0.06 was recorded. This is because pristine a-C:H surfaces are already passivated with hydrogen atoms. At the presence of saturated fatty acid, a SAM also forms. The friction coefficient of 0.06 for a-C:H is consistent with our results for ta-C lubricated with stearic acid (Fig. 1). On ta-C, saturated fatty acid can anchor and form a SAM. Since pristine ta-C surfaces possess a high density of dangling bonds, the presence of saturated fatty acid improves the friction coefficient significantly. We thus changed the expression “relatively high friction” to “low friction”, and added the following brief explanation about previous experiments of steel/steel and a-C:H/a-C:H contacts with saturated fatty acids and a comparison with ta-C/ta-C contacts to the discussion of the revised manuscript.

“Thus the adsorption of many stearic acid molecules can lead to the formation of SAMs that prevent cold-welding and mitigate friction to some extent. The effectiveness of saturated fatty acids for friction reduction is also observed in steel/steel and a-C:H/a-C:H experiments under boundary lubrication⁹. The formation of a close-packed SAM reduces the friction coefficient from 0.15 to 0.10 for steel-steel tribopairs. In a-C:H/a-C:H contacts, regardless of whether saturated fatty acids are present or not, friction coefficient of the order of 0.06 were recorded, in agreement with our results for stearic-acid-lubricated ta-C/ta-C pairs. In the presence of saturated fatty acids, a-C:H surfaces are also covered with a close-packed SAM. Atomistic SAM models predict a higher friction coefficient for elaidic and oleic acid than stearic acid due to their looser packing¹⁴⁻¹⁷. As a result, this is not a valid model for a super- and ultra-low friction regime of ta-C lubricated with unsaturated fatty acids.”

6) Reviewer’s comment:

Page 2: From their Raman data, the authors suggest that oleic acid 'mitigates sp³→sp² rehybridization'. This assertion seems to be in disagreement with surface analysis in ref 7, which suggested that superlow friction of DLC in presence of oleic acid could be due to 'triboformation of a graphene-like structure' (i.e. sp³→sp² at the surface). This important discrepancy needs to be discussed here as well as on page 4. Perhaps oleic acid promotes a small amount of graphitisation at the interface (ref 7) but prevents sp³→sp² transitions in the bulk substrate?

Answer:

The referee raised an interesting point. In Ref. 7 synchrotron XPS is used as extremely surface sensitive tool to measure the chemical structure of the top carbon layers. Conversely, Raman spectroscopy provides structural information that is averaged over the entire coating. Therefore, we agree with the reviewer’s interpretation of our Raman results that oleic acid could promote an aromatisation at the very interface while concurrently reducing rehybridization in the bulk of the coating. Additional support of the referee’s interpretation comes from our new QMD simulations of ta-C lubricated with glycerol that demonstrate the formation of a one monolayer thick graphene-like surface passivation. We discuss and clarify this important aspect in discussion part of the revised manuscript.

“This reduces sp³→sp² transitions in the bulk of the coating and restricts the rehybridization to the topmost carbon layers. When comparing this result to our experiments one has to keep in mind that Raman spectroscopy provides average structural information of the entire coating. Since an aromatic passivation of the outermost carbon layers can only be detected by synchrotron XPS⁶, our Raman results merely indicate that the superlow friction observed for unsaturated fatty acids and glycerol protects the bulk of the coating from elastic deformations.”

7) Reviewer’s comment:

Page 3: In the introduction would be helpful to the reader to mention other molecular simulations of lubricant additive mechanochemistry e.g. refs [B] for 'reactive' CMD and [C] for QMD. It might also be useful to mention other CMD studies of stearic and oleic acid on steel surfaces e.g refs [D,E].

Answer:

We noticed that we did not mention other computational studies on tribochemistry of lubricant molecules confined in nano-channels and friction behaviour of self-assembled fatty acid films. In Ref. [B], we found a similar argument about anchoring of allyl alcohol molecules on a silica surface. Their reactive MD showed that mechanically-induced distortion of anchored molecules accelerates rates of tribochemical reactions, and hydroxyl groups and C=C double bonds involve polymerization at the buried interface. In Ref. [C], mechanically-induced decomposition of phosphorus additives were presented by means of ab initio molecular dynamics.

In Refs. [D,E], friction properties of fatty acid self-assembled monolayers (SAMs) adsorbed on iron oxide were studied with non-reactive MD. Interestingly, stearic and oleic acid yield almost the same, relatively high friction coefficient (0.10–0.15). The friction coefficient for stearic acid is in agreement with our results, whereas that for oleic acid differs from our results. This indicates that the friction mechanism of oleic acid observed in this study is completely different from the SAM model, which highlights the importance of mechanochemical fragmentation and resulting surface passivation. Instead of citing Refs. [B-D] in the introduction we discuss these articles at various places in context with our findings.

8) Reviewer's comment:

Page 4: The authors declare that unsaturated carboxylic acids give lower friction because they 'can bridge the tribogap by chemisorption of their two reactive centres'. This is counter-intuitive since generally strong chemical bonding to both surfaces prevents sliding and increases friction. 'Bridging' probably means that the mechanical forces on molecules adsorbed to opposing counter-sliding surfaces are greater than if they are adsorbed to one of not adsorbed at all, leading to larger mechanochemical forces and thus faster passivation of the surface (as shown in Figure 3). This needs to be further explained here and in the results section (see e.g. ref [B] which suggests that adsorption is important for mechanochemical reactions to occur).

Answer:

The reviewer's comment makes us realise that we did not explain this clearly enough in the original manuscript. As the reviewer mentioned, bridging of confined lubricant molecules gives rise to larger mechanical forces than anchoring or non-bonded states. However, such high-friction states are instantaneous and only occur at the beginning of sliding, i.e. during running-in. The large shear forces induce strain in cross-linked molecules, leading to a cascade of molecular fragmentation reactions. This mechanically-induced fragmentation releases passivating hydrogen atoms and oxygen functional groups to ta-C surfaces. Figure 3c in the revised manuscript shows the evolution of the shear force F_F for a QMD trajectory of cis-3-heptenoic acid (shown in Fig. 3a). There are a few spikes of F_F (corresponding to cross-

linking), but the much larger shear forces of about 4 nN lead to a fast fragmentation and resulting surface passivation, which prevents further surface reactions. We added the following sentences to the result section of the revised manuscript.

“Although the cross-linking is accompanied by high friction during running-in, the related large shear forces induce tensile strain in the molecules. This causes their fragmentation and the resulting chemical passivation leads to nearly wearless sliding and superlow friction of ta-C⁵.”

In Ref. [B], the authors showed that when alcohol molecules anchor on one of the surfaces, distortion in the anchored molecules occurs and thus accelerates rates of polymerization reactions. Polymerization occurs at reactive sites (C=C bonds and OH groups). The similarities are worth noting to verify our results. We thus added the following sentences to the discussion of the revised manuscript.

“Similar tribochemical processes were observed in reactive atomistic simulations of the polymerization of benzene¹⁸ and allyl alcohols¹⁹ and decomposition of phosphorus additives²⁰. Yeon et al.¹⁹ suggested that distortions of anchored alcohols accelerate chemical reactions and polymerization occurs at double-bonded carbon and hydroxyl oxygen.”

9) Reviewer’s comment:

Page 4: It is noteworthy that the trend shown in Fig 1 is the opposite to that observed for organic friction modifiers in steel-steel contacts (see ref [F]). Specifically, for steel-steel, friction increases from: stearic acid (saturated tail) < elaidic acid (trans-unsaturated) < oleic acid (cis-unsaturated). This is because stearic acid forms close-packed monolayers more readily (see ref [G]) which has been shown in classical MD simulations to give lower friction (see ref [E]). On ta-C surfaces, close-packed monolayer formation is clearly less important to low friction than on steel (ref [A]).

Answer:

We absolutely agree with this comment. Our atomistic simulations also show that stearic acid anchors on ta-C surfaces at carboxylic polar head. The formation of a self-assembled monolayer (SAM) can reduce the friction and wear. This is the same as for steel/steel contacts (Refs. [E, G]). However, in the SAM model, stearic acid yields lower friction than elaidic and oleic acid since stearic acid forms closer-packed SAM (Ref. [E]). This is completely opposite to our results, which means that the SAM model is not valid for unsaturated fatty acid lubrication but our model is. Interestingly, in the SAM model the kink inhibits close packing, whereas in our model it facilitates cross-linking and mechano-chemical decomposition. The different lubrication mechanism of steel/steel and ta-C/ta-C pairs is worth a discussion, and thus a few sentences were added to the experiment section of the revised manuscript.

“Interestingly, atomistic SAM models predict a higher friction coefficient for elaidic and oleic acid than stearic acid due to their looser packing¹⁴⁻¹⁷. As a result, this is not a valid model for a super- and ultra-low friction regime of ta-C lubricated with unsaturated fatty acids.”

10) Reviewer’s comment:

Page 5: In addition to the shorter chain length, the functional groups also have closer proximity to each other in cis-3-heptenoic acid and trans-3-heptenoic acid compared to oleic acid and elaidic acid. This might also be an important factor and should be noted here.

Answer:

We think that this point is addressed by our new quantum-mechanical simulations of C₁₈ lubricants. The results show that the mechano-chemical decomposition mechanism is not affected by the proximity between COOH and C=C in unsaturated fatty acids. The critical pressures for anchoring and cross-linking are also similar. We added a section to the Supplementary Information of the revised manuscript.

“QMS simulations of C₁₈ fatty acids

In this study, due to huge computational efforts of quantum-mechanical simulations, we use C₇ model lubricants instead of C₁₈ lubricants. However, the QM modelling with C₁₈ fatty acids should be very instructive to rule out the effect of chain lengths and show the validity of our results. Therefore, we perform additional QMS simulations of C₁₈ fatty acids (stearic and oleic acid) confined between two ta-C blocks. For these simulations, a larger ta-C sample is created by expanding Sample 1 (used in Fig. 2 and 3) three times in the *x* direction. Figure S2 shows the same ordering in the critical pressures for anchoring and cross-linking as in Fig. 2. For oleic acid, a lower contact pressure of 2.83 GPa is required for cross-linking and it can anchor on one of the ta-C surfaces with almost no tribological loads. A pronounced kink and exposure of the C=C double bond to the counter surface are observed (middle panel in Fig. S2b). Although we generate only one trajectory for each, the result verifies the transferability of the C₇ result to C₁₈ molecules.“

In addition a short QMD trajectory of oleic acid showed a very similar mechanochemical decomposition as the cis-3-heptenoic acid (see new Fig. 3 b and c). We added a short description to the main text:

“Cross-linking and mechano-chemical fragmentation are also observed during a short QMD simulation with oleic acid (Fig. 3b), suggesting the transferability of the C₇ result to C₁₈ molecules.”

11) Reviewer’s comment:

Page 11: The authors suggest that the much lower friction in the cis isomer relative to the trans isomer are because they are 'more exposed to the counter body... due to its kink structure'. From the simulation results, can the authors be any more specific here to rationalise the large macroscopic differences?

Answer:

This comment inspired us to perform new QMS simulations for cis- and trans-3-heptenoic acid. In Fig. 2, we did not observe a clear ordering in the critical pressures for cross-linking. This is because chemical reactivities of the lubricants are more affected by local atomic configurations of ta-C surfaces than their chemical functionalities (COOH and C=C) due to strong heterogeneities of ta-C surfaces. In order to find a significant difference with reliable statistics, a large number of trajectories would be needed. Thus, we instead used a diamond (111) surface to rule out the heterogeneities of atomic configurations. Starting with an anchored state of an unsaturated fatty acid, we performed QMS pressurization simulations (Fig. S1 in Supplementary Information). In order to avoid the formation of other covalent bonds on the lower surface, the dangling bonds are fully terminated with hydrogen atoms. The results show a clear difference in the critical pressures for cross-linking. The averaged critical pressures of 0.44 ± 0.13 and 3.93 ± 1.36 GPa are obtained for cis and trans-3-heptenoic acid, respectively. This major difference is explained by different structural responses of the anchored molecules to pressure. Trans-3-heptenoic acid keeps the straight shape in the hydrocarbon chain and becomes parallel to the surfaces gradually as decreasing the surface separation. In contrast, the cis-3-heptenoic acid can bend at the C=C double bond site. The kink in the cis-3-heptenoic acid becomes more pronounced as decreasing the surface separation, which leads to the exposure of the double bond to the counter surface. The same pronounced bend is observed for oleic acid (Fig. S2 in Supplementary Information). Thus, the stereoisomerism effect is surely responsible for facilitating cross-linking and shear-induced fragmentation. This should be another criterion for designing lubricant additives. We think that the stereochemistry aspect is interesting and novel, but not a main finding of this study and needs a further investigation. Therefore, these results were added to Supplementary Information and a hint was given in the revised manuscript.

Revised main text:

“In particular, the more pronounced kink in cis-3-heptenoic acid represents a steric promoter for cross-linking (see Supplementary Information for the details), leading to a fast fragmentation into passivating functional groups.”

Addition to Supplementary Information:

“Stereoisomerism effect

Sliding tests of ta-C/ta-C pairs show a clear reproducible trend in the friction coefficient and wear between elaidic and oleic acid (Fig. 1): oleic acid lubrication yields a superlubricity regime ($\mu = 0.008$), whereas elaidic acid lubrication gives a slightly larger friction coefficient ($\mu = 0.017$). However, our QMS simulations do not provide a significant difference in the critical pressures for anchoring and cross-linking between trans- and cis-3-heptenoic acid (Fig.

2). Since ta-C surfaces are not homogeneous, surface chemistry strongly depends on local atomic configurations and thus a numerous number of trajectories have to be generated to obtain a significant difference as well as stable statistics. Therefore, instead of ta-C surfaces, we here use a non-reconstructed diamond (111) surface to rule out the impact of the local atomic configurations. An unsaturated fatty acid (trans- or cis-3-heptenoic acid) initially chemisorbs on one of the diamond (111) surfaces via carboxylic polar head. The lower surface is fully passivated with hydrogen atoms, but the upper one is not. Figure S1 shows different structural responses of the anchored molecules to normal pressure. Trans-3-heptenoic acid keeps the straight shape in the hydrocarbon chain and becomes parallel to the surfaces gradually as decreasing the surface separation h . In contrast, the cis-3-heptenoic acid can bend at the C=C double bond site. The kink in the cis-3-heptenoic acid becomes more pronounced as decreasing the surface separation h , which leads to the exposure of the double bond to the counter surface. The same pronounced bend is observed for oleic acid (Fig. S2). We generate five trajectories for each molecule, and the averaged critical pressures of 0.44 ± 0.13 and 3.93 ± 1.36 GPa are obtained for cis and trans-3-heptenoic acid, respectively. These results indicate that the stereoisomerism effect is surely responsible for facilitating cross-linking and shear-induced fragmentation.”

12) Reviewer’s comment:

Page 14: In addition to GMO and glycerol, could also potentially explain results of ref [H], where superlow friction observed with DLC lubricated by 1,3-Diketone (i.e. two ketone groups).

Answer:

This is an interesting idea. Our manuscript shows that the multiple-reactive-centre-driven mechano-chemical decomposition can be applied to ta-C lubricated with other common oxygen-containing lubricants (such as polyols). However, the authors in Ref. [H] used a-C:H (including 30 at% hydrogen) and assumed that the tribological systems are initially in the mixed lubrication regime and later in the hydrodynamic regime. The discrepancies make the direct comparison difficult. Moreover, although it is true that 1,3-Diketone possesses multiple reactive centres, the complicated structure might require a higher contact pressure for anchoring and cross-linking. Perhaps stereochemical aspects are important. Thus, we think that the prediction for 1,3-Diketone is out of the scope of this study.

13) Reviewer’s comment:

Page 17: Is the 5 GPa assuming complete contact of the (atomically smooth) slabs? If so, important to indicate that the 'local' pressure at the sinusoidal 'asperities' will be much higher than this.

Answer:

The local pressure at the sinusoidal asperity is of the order of the applied pressure (5 GPa). Since the lubricant completely fills the volume around the asperity underneath the upper

surface, the applied pressure is not simply localized at the asperity peak, but it is rather well distributed over the asperity and over the fluid lubricant molecules (octadecane or octadecene). A note was added to the revised manuscript.

“The local pressure at the sinusoidal asperity is almost as large as the applied pressure of 5 GPa. Since the liquid lubricants (octadecane or octadecene) fill the volume around the asperity underneath the upper surface, the applied pressure is well distributed over the asperity and over the fluid lubricant molecules.”

14) References suggested by the reviewer:

[A] Simič and Kalin, *Appl. Surf. Sci.*, 2013, 283, 460-470. [B] Yeon et al. *ACS Appl. Mater. Interfaces* 2017, 9, 3142–3148. [C] Loehle et al. *Lubricants* 2018, 6, 31. [D] Doig et al. *Langmuir*, 2014, 30, 186-195. [E] Ewen et al. *Langmuir*, 2016, 32, 4450-4463. [F] Campen et al. *Tribol. Lett.*, 2012, 48, 237-248. [G] Wood et al. *Langmuir*, 2016, 32, 534-540. [H] Amann et al. *ACS Omega*, 2017, 2, 8330-8342.

Answer:

Thank you for suggesting the interesting papers. We added them to the references of the revised manuscript (except Ref. [H] – see answer to comment 12)).

In the following we provide a point-by-point response to the comments of the reviewers. Revisions are marked in blue in this response letter and the revised manuscript.

Reviewer #1:

1) Reviewer's comment:

The authors have greatly improved the tribological part with including the glycerol case and calculations of EHL film thicknesses. However, the tests are made with very unusual tribological parameters (relatively high sliding speed, large radius of curvature of the pin and small kinematic length on the flat). The test resembles more a fretting one with the existence of Mindlin zone at the periphery of the pin wear scar. I was very suspicious that these specific conditions lead to the boundary regime in all cases, as stated by the authors. By using the tribological parameters indicated in the related part and a well-known calculator available on the website (http://www.tribology-abc.com/calculators/c4_10.htm), I found that the results

do not match with the authors' ones. I also used another in house calculator in the Lab that confirmed the previous one.

Answer:

We thank the reviewer for carefully reading our revised manuscript and for the effort she/he spent in trying to reproduce our EHL film thickness calculations. We agree with the reviewer that the SRV test was originally designed to quantify the performance of lubricants under fretting-like conditions (e.g., as specified in standard DIN 51834-2). However, the tribometer is capable to precisely measure low friction in high-frequency oscillating motion, as demonstrated in investigations of ultra- and super-low friction of ta-C coatings [Kano et al., Tribol. Lett. 18, 245 (2005) and De Barros Bouchet et al., J. Phys.: Conf. Ser. 89, 012003 (2007)]. In both works, the authors chose 50 Hz and 3 mm stroke length, resulting in about 0.3 m/s sliding speed at 270 MPa maximum Hertzian contact pressure. While these studies used a cylinder on flat geometry, we worked with curved pin on flat, using a large pin radius to obtain a larger contact area. In our experiments we have a maximum sliding speed of 0.157 m/s and a 240 MPa maximum Hertzian contact pressure, i.e. we have conditions that are similar to those in the references. We consciously extended the stroke from about 1 mm, often used in SRV experiments, to 5 mm in order to compensate the relatively large contact area of about 0.8 mm in diameter. Thereby, the sliding length is much longer than the contact zone, avoiding a fretting situation and generating a pronounced sliding region between the reversing points.

We are also very grateful to the reviewer for suggesting the useful website, where EHL film thickness calculations are based on a Nijebanning-Venner-Moes equation [Nijebanning et al., Wear 176, 217 (1994)]. Wheeler et al. performed a comparison of the EHL film thickness using different equations, and showed that the Nijebanning-Venner-Moes equation is the most accurate in reproducing the numerical solutions of the basic EHL equations [Wheeler et al., Tribol. Lett. 4, 369 (2016)]. Thus, we also calculated film thickness values using the Nijebanning-Venner-Moes equation, but found that the calculation on the website gives the central film thickness h_{centre} (i.e., the film thickness below the centre of the pin). In general, the minimum film thickness h_{min} rather than h_{centre} should be used for the calculation of λ (defined as $\lambda = \frac{h_{\text{min}}}{\sqrt{R_{\text{q,pin}}^2 + R_{\text{q,disc}}^2}}$, where $R_{\text{q,pin}}$ and $R_{\text{q,disc}}$ are the root-mean-square (RMS)

values of the surface roughness of the spherical pin and disc, respectively). As a result, the lambda values suggested by the reviewer should be much larger than the correct ones. In the Nijebanning-Venner-Moes equation, a minimum film thickness h_{min} can be estimated using an empirical $h_{\text{centre}}/h_{\text{min}}$ parameter [Wheeler et al., Tribol. Lett. 4, 369 (2016)]. In addition, since surface roughness values are crucial in the estimation of λ and since the previous value of 20 nm was a rough estimate, we performed more accurate measurements of the RMS roughness of our surfaces. The new measurements turned out to be crucial as, for instance, RMS roughness values R_{q} of 37.1 nm (pin) and 43.3 nm (disc) were obtained after the glycerol test. With these roughness values, a lambda value smaller than one ($\lambda = 0.87$) is obtained also for glycerol. The lambda values for the other lubricants are even smaller. This clearly suggests that our systems slide in boundary lubrication conditions for all lubricants. Note, that we now use the maximum sliding velocity ($v_{\text{max}} = 0.157$ m/s) instead of the

average in our new calculations, since we wanted to ensure that we are in boundary lubrication throughout a complete friction cycle.

A “weaker” indication of the fact that our measurements are in the boundary lubrication regime was provided by Reviewer #3 who pointed out that, in the reciprocating tribometer, at the end of each stroke, the sliding speed goes to zero, resulting in a significant decrease of the fluid film thickness and an increase of asperity-asperity contacts, ensuring that a boundary lubrication regime is established at least near the reversing points of the strokes. Our new calculations of the EHL film thickness extend the boundary lubrication conditions to the rest of the pin trajectory. As a result, it is safe to state that the frictional and wear behaviour of our tribosystems is generally not affected by lubricant viscosities. Indeed, our experiments show that there is no clear correlation between the friction coefficients and viscosities. In particular, the comparison between stearic and oleic acid is the most instructive. The minimum EHL film thickness for stearic acid is larger than that for oleic acid, but stearic acid yields much higher friction and wear. In the Supplementary Information, we replaced the Hamrock-Dowson results with Nijebanning-Venner-Moes results, and added a brief discussion about the reciprocating tester and actual friction regime.

2) Reviewer’s comment:

For the glycerol case, I found the max pressure of 240 MPa, a radius of contact area of 314 microns (in agreement with the optical images in Fig 1) and a film thickness of 56 nm (see screen shot attached). Lambda is about 3 against 0.86 by the authors. For me, EHL is the main regime in this particular experiment.

Answer:

We reproduced the reviewer’s result using the calculator on the suggested website. Indeed, we obtained a film thickness of 56 nm for glycerol. However, as mentioned above, the website gives a central film thickness using a Nijebanning-Venner-Moes equation. Perhaps, the reviewer may have used $\lambda = \frac{h_{\text{centre}}}{R_q}$ rather than the formula above which uses the minimum value of the film thickness and the combined roughness value. Using this formula, our λ estimation yields a value $\lambda = 0.87$, which suggests that, even for glycerol, the tribological systems are in the boundary lubrication regime. Moreover, as explained above, boundary lubrication is surely achieved at the reversing points of each stroke. A similar SRV reciprocating test of ta-C with glycerol was also run under boundary lubrication by Matta and coworkers [Phys. Rev. B 78, 085436 (2008)].

3) Reviewer’s comment:

For oleic acid, and according to the same calculator, the film thickness is about 21 nm. So lambda value is unity (against 0.3 in the manuscript). Moreover one must take into account the effect of elastic deformation of asperities in the contact area that has the effect to decrease the roughness although this is not well documented in the literature. Again these are not boundary lubrication conditions but rather mixed conditions so that a large part of the friction is ensured by viscosity properties of the acid and not by contacting solid asperities where simulations are made.

Answer:

As stated in our answers above, the reviewer's result of $h = 21$ nm for oleic acid is based on a Nijebanning-Venner-Moes equation. However, this value is a central film thickness while a minimum film thickness is required for the estimation of λ . We estimate $h_{\min} = 15.64$ nm and $\lambda = 0.51$ for oleic acid using the $h_{\min}/h_{\text{centre}}$ ratios tabulated in Wheeler et al., Tribol. Lett. 4, 369 (2016).

The reviewer comment on the roughness reduction due to the elastic deformation of asperities is very helpful. To check the effect of elastic deformation on the surface roughness in the contact area, we considered a model spherical asperity with radius of 20 nm (as suggested by the roughness) and a typical local pressure of 5 GPa. According to the Hertz contact theory for elastic half-space, the penetration depth d is defined by $d = R \left(\frac{3\pi P}{4E'} \right)^{\frac{2}{3}}$, where R is the radius of curvature of the asperity and P is the normal pressure. Using $R = 20$ nm and $P = 5$ GPa results in $d = 1.78$ nm. Since the Young's modulus of ta-C is much larger than P , the elastic deformation does not significantly decrease the surface roughness of the asperities. Thus, even after taking into account the elastic deformation, the λ values are still well below unity. We added a brief note about the elastic deformation of asperities and resulting decrease of the surface roughness in the Supplementary Information.

4) Reviewer's comment:

For the hydrocarbon, the film thickness is well below the roughness value (9 nm) and the regime is boundary with corresponding high friction.

Answer:

The reviewer is right. The smallest film thickness is obtained for octadecane. However, $h = 9$ nm on the website is a central film thickness. We obtained a minimum film thickness h_{\min} of 5.41 nm for octadecane.

5) Reviewer's comment:

So, unless an error has happened in the data related in the manuscript, the regimes are not the same for the different lubricants and this is a pity (as I already mentioned in my first round of comments).

So my opinion is that the paper in the present state is not acceptable from the tribological point of view. To be acceptable, the glycerol case should be removed and the oleic acid lubrication test should be stated in the mixed regime (or transition mixed/boundary). Even in this case it is not sure that the friction supported by asperities is a major part compared with the oleic acid nanofilm between asperities.

Answer:

As explained above, our film thickness calculations show that for all lubricants the systems are under boundary lubrication.

6) Reviewer's comment:

I would suggest that the authors decrease the sliding speed by a factor three for example, thus making all tests to run in boundary conditions where computer simulations are valid. However it is not sure that superlubricity is retained in this condition.

Answer:

Since our new EHL calculations of the film thickness, result in lambda parameters that are smaller than one in all cases presented in the manuscript even at a maximum sliding speed v_{\max} of 0.157 m/s, we decided not to perform additional experiments at a lower sliding speed.

Reviewer #2:

1) Reviewer's comment:

The authors have addressed the concerns raised. While some limitations (speed, molecule length) are not possible to overcome due to computational limitations, trends and other reasonable arguments are presented that support the hypotheses of this work. This paper is a notable advance in providing ideas and knowledge that can contribute to the rational design of lubricants, which itself is increasingly important given the critical need to reduce energy consumption and wear in engineered systems. I believe it is not an exaggeration to say that rational tribology needs to be a crucial part of saving the planet from the threat of excessive CO2 emissions, and this work and work like it will help form the basis for such efforts. I strongly recommend publication.

Answer:

We are grateful to the reviewer for his very positive assessment of our manuscript and for supporting its publication in *Nature Communications*.

Reviewer #3:

1) Reviewer's comment:

I should start by thanking the authors for their comprehensive response to my comments. Overall, I think that the manuscript has been greatly improved following the extensive revisions. The arguments presented are now well-supported by more compelling simulation results. I have some remaining minor concerns which should be addressed prior to publication.

Answer:

We thank the reviewer for the positive assessment of our revised manuscript. We have further improved the manuscript according to the reviewer's comments and suggestions.

2) Reviewer's comment:

Page 2, line 33 – perhaps helpful to cite review of DLC in automotive industry e.g. A. Erdemir. Tribol. Int. 2004, 37, 577-583

Answer:

We thank the reviewer for suggesting the important publication, which we now cite in the manuscript.

3) Reviewer's comment:

Page 2, line 36 – need to outline that SRVs are high-frequency reciprocating tribometers (for non-tribology audience)

Answer:

We replaced “SRV experiments” with “reciprocating sliding tests”.

4) Reviewer's comment:

Page 5, line 136 – should state good agreement with previous macroscopic tribology experiments on the same systems e.g. refs [4-6].

Answer:

We added the following sentence: “in good agreement with previous experiments on the similar systems⁵⁻⁷”.

5) Reviewer's comment:

Page 14, line 291 – typo: ‘less reactive alkyl rest does not’ to read ‘unreactive, saturated alkyl tail does not’

Answer:

We corrected it.

6) Reviewer's comment:

Page 16, line 340 – more context needed: ‘can arrange in a close-packed SAM14,15, the softer carpet’ to read ‘can arrange in a close-packed SAM14,15, relative to ta-C, the softer carpet’

Answer:

We corrected it.

7) Reviewer's comment:

Page 17, line 382 and Supporting Information – Film thickness calculations from the Hamrock-Dowson equation, designed for non-conformal EHL contacts, have limited applicability here. This is a reciprocating pin-on-disk tribometer (SRV), where there is zero entrainment speed at the ends of each stroke. Thus at the ends of each stroke, the system will be under boundary conditions (see e.g. Graham et al. Tribol. Trans. 44, 626-636, 2001). It is noteworthy that such tribometers were originally developed as a wear test (Wei et al. Wear. 111, 217-235, 1986) and significant wear is observed for some of the fluids in the current manuscript. I strongly suggest removing the EHL film thickness calculations and simply state that tests are under boundary conditions where load is supported mostly by surface asperities and lubricant viscosity has negligible effect on friction and wear, citing the above references in the main text.

Answer:

We thank the reviewer for the constructive comment and for suggesting some interesting references. We added the description about the zero entrainment speed at the reversing points to the revised manuscript.

We reconsidered the EHL film thickness calculations and found that the applicability of these estimations are already validated for similar SRV reciprocating tests [e.g. Okubo et al., *Wear* 332, 1293 (2015)] and that a Nijenbanning-Venner-Moes equation is more accurate than the Hamrock-Dowson equation [Nijenbanning et al., *Wear* 176, 217 (1994) and Wheeler et al., *Tribol. Lett.* 4, 369 (2016)]. We considered removing the calculation of the film thickness, as suggested by the reviewer. However, the estimate of the film thickness is a crucial point that we need to report in order to answer the concerns of Reviewer #1. Moreover, we believe that the film thickness calculation is useful to show that the systems are under boundary lubrication everywhere in the wear track. Thus, we removed the previous Hamrock-Dowson estimations and instead added the Nijenbanning-Venner-Moes calculations.

8) Reviewer's comment:

Page 21, line 497 – citations for LAMMPS (S. Plimpton. *J. Comput. Phys.* 117, 1-19, 1995) OVITO (A. Stukowski, *Modell. Simul. Mater. Sci. Eng.* 18, 015012, 2010) and VMD (W. Humphrey et al. *J. Mol. Graphics Modell.* 14, 33-38, 1996) should be added.

Answer:

Thanks for pointing out these references. We added them to the manuscript.

Reviewers' comments:

Reviewer #1 (Remarks to the Author):

The authors have greatly improved the tribological part with including the glycerol case and calculations of EHL film thicknesses.

However, the tests are made with very unusual tribological parameters (relatively high sliding speed, large radius of curvature of the pin and small kinematic length on the flat). The test resembles more a fretting one with the existence of Mindlin zone at the periphery of the pin wear scar.

I was very suspicious that these specific conditions lead to the boundary regime in all cases, as stated by the authors. By using the tribological parameters indicated in the related part and a well-known calculator available on the website (http://www.tribology-abc.com/calculators/c4_10.htm), I found that the results do not match with the authors' ones. I also used another in house calculator in the Lab that confirmed the previous one.

For the glycerol case, I found the max pressure of 240 MPa, a radius of contact area of 314 microns (in agreement with the optical images in Fig 1) and a film thickness of 56 nm (see screen shot attached). Lambda is about 3 against 0.86 by the authors. For me, EHL is the main regime in this particular experiment.

For oleic acid, and according to the same calculator, the film thickness is about 21 nm So lambda value is unity (against 0.3 in the manuscript). Moreover one must take into account the effect of elastic deformation of asperities in the contact area that has the effect to decrease the roughness although this is not well documented in the literature. Again these are not boundary lubrication conditions but rather mixed conditions so that a large part of the friction is ensured by viscosity properties of the acid and not by contacting solid asperities where simulations are made.

For the hydrocarbon, the film thickness is well below the roughness value (9 nm) and the regime is boundary with corresponding high friction.

So, unless an error has happened in the data related in the manuscript, the regimes are not the same for the different lubricants and this is a pity (as I already mentioned in my first round of comments).

So my opinion is that the paper in the present state is not acceptable from the tribological point of view. To be acceptable, the glycerol case should be removed and the oleic acid lubrication test should be stated in the mixed regime (or transition mixed/boundary). Even in this case it is not sure that the friction supported by asperities is a major part compared with the oleic acid nanofilm between asperities.

I would suggest that the authors decrease the sliding speed by a factor three for example, thus making all tests to run in boundary conditions where computer simulations are valid. However it is not sure that superlubricity is retained in this condition.

Reviewer #2 (Remarks to the Author):

The authors have addressed the concerns raised. While some limitations (speed, molecule length) are not possible to overcome due to computational limitations, trends and other reasonable arguments are presented that support the hypotheses of this work. This paper is a notable advance in providing ideas and knowledge that can contribute to the rational design of lubricants, which itself is increasingly important given the critical need to reduce energy consumption and

wear in engineered systems. I believe it is not an exaggeration to say that rational tribology needs to be a crucial part of saving the planet from the threat of excessive CO₂ emissions, and this work and work like it will help form the basis for such efforts. I strongly recommend publication.

Reviewer #3 (Remarks to the Author):

Reviewer Comments for Revised Manuscript

I should start by thanking the authors for their comprehensive response to my comments. Overall, I think that the manuscript has been greatly improved following the extensive revisions. The arguments presented are now well-supported by more compelling simulation results. I have some remaining minor concerns which should be addressed prior to publication.

Page 2, line 33 – perhaps helpful to cite review of DLC in automotive industry e.g. A. Erdemir. Tribol. Int. 2004, 37, 577-583

Page 2, line 36 – need to outline that SRVs are high-frequency reciprocating tribometers (for non-tribology audience)

Page 5, line 136 – should state good agreement with previous macroscopic tribology experiments on the same systems e.g. refs [4-6].

Page 14, line 291 – typo: 'less reactive alkyl rest does not' to read 'unreactive, saturated alkyl tail does not'

Page 16, line 340 – more context needed: 'can arrange in a close-packed SAM_{14,15}, the softer carpet' to read 'can arrange in a close-packed SAM_{14,15}, relative to ta-C, the softer carpet'

Page 17, line 382 and Supporting Information – Film thickness calculations from the Hamrock-Dowson equation, designed for non-conformal EHL contacts, have limited applicability here. This is a reciprocating pin-on-disk tribometer (SRV), where there is zero entrainment speed at the ends of each stroke. Thus at the ends of each stroke, the system will be under boundary conditions (see e.g. Graham et al. Tribol. Trans. 44, 626-636, 2001). It is noteworthy that such tribometers were originally developed as a wear test (Wei et al. Wear. 111, 217-235, 1986) and significant wear is observed for some of the fluids in the current manuscript. I strongly suggest removing the EHL film thickness calculations and simply state that tests are under boundary conditions where load is supported mostly by surface asperities and lubricant viscosity has negligible effect on friction and wear, citing the above references in the main text.

Page 21, line 497 – citations for LAMMPS (S. Plimpton. J. Comput. Phys. 117, 1-19, 1995) OVITO (A. Stukowski, Modell. Simul. Mater. Sci. Eng. 18, 015012, 2010) and VMD (W. Humphrey et al. J. Mol. Graphics Modell. 14, 33-38, 1996) should be added.

Reviewers' comments:

Reviewer #1 (Remarks to the Author):

The authors have tried to justify that the different lubricated tests are performed in the boundary lubrication regime (λ ratio below unity).

However I does not agree with these comments for the following reasons:

1- The authors used h_{min} to calculate the λ ratio. This is not appropriate to discuss the friction properties because the area with h_{min} is a few per cent of the total Hertzian zone and corresponds to a very low contact pressure (see white area in image attached). The h_{min} is only used to discuss on possible damage at the periphery of the contact area and contribution to friction force is completely negligible.

2- - There is a mistake in the calculation of λ for the glycerol case. Even when using h_{min} , the λ ratio is 1.2. It becomes between 2 and 3 when using h_{centre} . So this case is not in the boundary regime. It is also slightly above unity for the oleic acid case when using h_{centre} .

3- Why did the authors use rough ta-C surfaces when they can polished easily ?

4- Because the frequency of the friction machine is high, it is not sure that the oil film has the time to be squeezed at the zero speed point. This is because of the so-called squeeze film. To be sure of that, the friction force over one cycle should be observed to distinguish the CoF at the changing direction and in the middle of the stroke.

So my opinion is that the glycerol case should be removed because simulation is not appropriate to EHL lubrication. Even the rest of the paper with oleic acid is questionable in my opinion..
New tribological tests in the boundary regime are needed.

Reviewer #2 (Remarks to the Author):

I have reviewed the responses of the authors to all referees, paying close attention to the comments of the first referee. The authors are correct on this point, that h_{min} is the proper thickness to choose. This resolves all of the concerns of referee #1. The papers has been improved further thanks to this feedback and other feedback. I recommend publication.

Reviewer #3 (Remarks to the Author):

No further comments.

In the following we provide a point-by-point response to the comments of the reviewers. Revisions are marked in blue in the revised Supplementary Information.

Reviewer #1:

1) Reviewer's comment:

The authors used h_{\min} to calculate the lambda ratio. This is not appropriate to discuss the friction properties because the area with h_{\min} is a few percent of the total Hertzian zone and corresponds to a very low contact pressure (see white area in image attached). The h_{\min} is only used to discuss possible damage at the periphery of the contact area and contribution to friction force is completely negligible.

Answer:

While h_{\min} is very often used for the lambda estimation in many similar studies (as Reviewer #2 agreed), sometimes the application of a somehow vaguely defined average film thickness in the flat part of the contact is discussed. Maybe the latter could be defined by h_{centre} (λ_{centre}). We believe that the use of h_{\min} (λ_{\min}) should be more appropriate for our tribosystems for the following reason.

As earlier suggested by Reviewer #1, we performed additional sliding tests of glycerol with lower reciprocating frequencies ($f = 4$ and 5 Hz). For these frequencies, λ is well below unity even when using h_{centre} . At lower frequency, the same superlow friction coefficient ($\mu = 0.008$) was recorded as at $f = 10$ Hz (shown in Fig. 1). This result suggests that also our initial experiments at $f = 10$ Hz were carried out in the boundary lubrication regime. Since at $f = 10$ Hz, $\lambda_{\min} = 0.87$ and $\lambda_{\text{centre}} = 1.30$ are obtained, we conclude that λ_{\min} more consistently reflects the tribological behaviour actually observed in the original experiments.

2) Reviewer's comment:

There is a mistake in the calculation of lambda for the glycerol case. Even when using h_{\min} , the lambda ratio is 1.2. It becomes between 2 and 3 when using h_{centre} . So this case is not in the boundary regime. It is also slightly above unity for the oleic acid case when using h_{centre} .

Answer:

We greatly appreciate the effort that the reviewer spent in trying to reproduce our lambda calculations. We checked if our lambda calculations are correct and did not find any mistake. Perhaps, the reviewer used a wrong definition for the lambda value. For instance, when using $\lambda = h_{\min} / \frac{R_{q,\text{pin}} + R_{q,\text{disc}}}{2}$, we obtained $\lambda = 1.23$ for glycerol. For h_{centre} , it gives $\lambda = 1.85$ and 1.28 for glycerol and oleic acid, respectively. These values are close to the numbers suggested by the reviewer. However, the use of the arithmetic mean of the roughnesses is not correct. The denominator is defined as a composite roughness $\sqrt{R_{q,\text{pin}}^2 + R_{q,\text{disc}}^2}$ [Tallian, ASLE Trans. 10, 418 (1967)]. Therefore, we are sure that our lambda ratios reported in Table S2 are correct.

3) Reviewer's comment:

Why did the authors use rough ta-C surfaces when they can be polished easily?

Answer:

The numbers reported in Table S1 are not the surface roughness values after polishing. We measured the surface roughness of ta-C surfaces after running-in since we consider them more meaningful for the steady-state lambda values.

4) Reviewer's comment:

Because the frequency of the friction machine is high, it is not sure that the oil film has the time to be squeezed at the zero speed point. This is because of the so-called squeeze film. To be sure of that, the friction force over one cycle should be observed to distinguish the CoF at the changing direction and in the middle of the stroke.

Answer:

We operated the Optimol SRV tester already close to its low frequency limit. Of course, a squeeze film could still result in finite lambda values at the end of the stroke. According to Persson's review [J. Phys.: Condens. Matter 16, R295 (2004)], the squeeze-out dynamics of a fluid film from the gap can be approximately described by $\frac{1}{h^2(t)} - \frac{1}{h^2(0)} = \frac{4tP}{3\eta R^2}$, where P is the mean Hertzian contact pressure, η is the fluid viscosity and R is the Hertzian contact radius. Since $h(t) \sim t^{-1/2}$, the actual λ should be smaller than the maximum estimate ($\lambda = 0.87$, Table S2). Indeed, assuming $t = \frac{1}{4f}$ the minimum film thickness and the corresponding lambda values turn out to be $h_{min} = 29.9$ nm and $\lambda = 0.52$, respectively. So, for $f = 10$ Hz, this simple estimate suggests that at the end of the stroke the film thickness and lambda are not zero, but they are significantly reduced and thus the test is performed in boundary lubrication. This is confirmed by our additional experiments of glycerol with lower reciprocating frequencies ($f = 4$ and 5 Hz resulting in $\lambda < 1$ over the whole stroke) that yield exactly the same superlow friction coefficient.

5) Reviewer's comment:

So my opinion is that the glycerol case should be removed because simulation is not appropriate to EHL lubrication. Even the rest of the paper with oleic acid is questionable in my opinion. New tribological tests in the boundary regime are needed.

Answer:

As a result of the reviewer's comment, we performed additional sliding tests of oleic acid and glycerol with lower frequencies f . For glycerol, at $f = 4$ and 5 Hz, λ decreases to $\lambda = 0.49$ and 0.57 , respectively. Even when the central film thickness h_{centre} is used, λ is well below unity ($\lambda = 0.74$ at $f = 4$ Hz). Thus, these new experiments were clearly performed under boundary lubrication. Importantly, the steady-state friction coefficients at these smaller frequencies are the same as at $f = 10$ Hz and $\lambda = 0.87$ ($\mu = 0.008$, Fig. 1). Also for oleic acid, at $f = 5$ Hz ($\lambda = 0.33$) the friction coefficient remains on a superlubricity level. These results provide strong support that also our earlier experiments with $f = 10$ Hz were carried out under boundary lubrication. We added a brief explanation about the new experiments for glycerol at lower frequencies to the Supplementary Information.

Reviewer #2:

1) Reviewer's comment:

I have reviewed the responses of the authors to all referees, paying close attention to the comments of the first referee. The authors are correct on this point, that h_{\min} is the proper thickness to choose. This resolves all of the concerns of referee #1. The paper has been improved further thanks to this feedback and other feedback. I recommend publication.

Answer:

We are grateful to the reviewer for carefully reading our manuscript and recommending publication of the revised manuscript in *Nature Communications*.

REVIEWERS' COMMENTS:

Reviewer #1 (Remarks to the Author):

I can understand that that the authors want to insist to prove that they are running tests under pure boundary lubrication in order to match with their computer simulations results between solid-like contacts in presence of molecules. In spite of their efforts, I don't think they can convince the tribology community because their friction results are in the borderline between EHL and mixed lubrication where fluid film effects are still very significant. It would have been very easy and appropriate here to run their tests in pure boundary lubrication by decreasing the sliding speed by a factor ten, for example. Maybe the two main parts of this paper have been obtained independently and put together afterwards. Nevertheless, should the editor decide to publish this paper, I propose the following mandatory revisions:

Concerning glycerol case:

The authors cite Matta's work in PRB 2008. However I found that there is another more detailed paper from Matta in 2008 (published in Lubrication Science). In this paper, the authors studied ta-C/ta-C tribopairs in glycerol at 80 °C at very low sliding speed (λ ratio below 0.1 in pure BL conditions). The CoF is about 0.02-0.04. Also they ran tests in glycerol vapour phase at same temperature. The CoF is about 0.05 in well agreement with results in Fig. 8 of submitted paper which matches very well the experimental conditions of the gas phase. So the results in this reference are in strong contradiction with the submitted paper with CoF 5 times higher. The authors must cite this "Lub. Science" paper and must explain why their results are in discrepancy with literature data.

Concerning oleic acid case:

The authors cite Kano's work with oleic acid. In this paper the Stribeck curve is shown and this is the proper way to define accurately the lubrication regime. It is clearly shown in Kano's work that ta-C/ta-C in glycerol does not display superlubricity under boundary lubrication (at very low speeds). This is again in strong contradiction with the submitted paper where superlubricity is said to happen under BL. The authors must comment and explain why they did not define the Stribeck curve to justify the lubrication regime.

Reviewer #4 (Remarks to the Author):

Some simple calculations allow one to demonstrate that the lubricated contacts in this paper are indeed operating in the boundary regime. Using Hertzian theory for point contacts, one can calculate that the maximum contact pressure is 0.23 GPa, the contact radius is 0.00032 m and the contact area is $0.32 \times 10^{-6} \text{ m}^2$. From the viscosity and pressure-viscosity coefficients given in Table S2, the viscosities at 0.23 GPa pressure can be determined to be: octadecane 15 mPa-s, stearic acid 38 mPa-s, oleic acid 26 mPa-s and glycerol 110 mPa-s.

With these relatively small viscosity increases due to pressure, the traction (friction) coefficient, if the contact were in a high λ value hydrodynamic regime, can be found from $\mu = (\text{viscosity} \times \text{area} \times \text{velocity}) / (\text{film thickness} \times \text{load})$. For the four oils above, μ (hydrodynamic) is 0.0023, 0.0020, 0.0021 and 0.0029, respectively. The values the authors measured in their pin on disk rig were 0.0700 (short period of steady state before what looks like scuffing), 0.0620, 0.0080 and 0.0080 (extra zeros are added for easier visual comparison with the calculated results).

It is possible to subtract the hydrodynamic friction to determine the boundary friction contribution. This indicates that all four lubricants had a significant friction contribution from asperity interactions, and therefore operated in the boundary lubrication regime. For octadecane and stearic acid, the boundary friction contribution was 30 times that of the hydrodynamic

contribution. For oleic acid and glycerol it was only 2-3 times, suggesting that these are good boundary friction reducers.

Note that pure hydrodynamic films at $\mu \sim 0.002$ are already in the range of super-lubricity and that the crankshaft, connecting rod and camshaft bearings and much of the piston-ring cylinder stroke is already operating in this very low friction regime with full hydrodynamic films. To argue in the introduction to the paper that reducing μ to <0.01 would improve fuel economy by 61% is a great exaggeration and hurts the credibility of the paper. The Erdimir and Holmberg paper is widely misused and misquoted.

With regard to reviewer 1's comments about which film thickness to use, I agree that it should be the central value not the minimum value when discussing friction, since that is where the majority of the load is carried. However, determining a λ value is tricky. The roughness value is a function of the length of surface measured. Sayles** found that over many length scales the rms roughness increased with the square root of the length measured. An appropriate length in tribology is the width of the contact, which in this case is 640 μm (see paragraph 1 above). The authors measured a 50 μm length with the AFM so they should multiply their rms values by $(640/50)^{0.5}$ or 3.6. Doing this would lower all their λ values by 3.6x and so they are all clearly in the boundary regime, regardless of whether minimum or central film thickness is used.

The authors could have checked the hydrodynamic contribution by running steel on steel pin on disc experiments with superfinished surfaces (5 nm rms) to confirm that their "rough" ta-C coatings are running in the boundary regime. However, I don't think that is necessary for publication.

In table S2, "curvature" should be "radius of curvature" as curvature is $1/\text{radius}$ and has units of m^{-1} .

**Sayles R.S., "The Topography of Surfaces", PhD Thesis, Teesside Polytechnic, 1976.

Response to Reviewer #1 and #4

In the following we provide a point-by-point response to the comments of the reviewers. Revisions are marked in blue in the revised manuscript and Supplementary Information.

Reviewer #1:

1) Reviewer's comment:

I can understand that that the authors want to insist to prove that they are running tests under pure boundary lubrication in order to match with their computer simulations results between solid-like contacts in presence of molecules. In spite of their efforts, I don't think they can convince the tribology community because their friction results are in the borderline between EHL and mixed lubrication where fluid film effects are still very significant. It would have been very easy and appropriate here to run their tests in pure boundary lubrication by decreasing the sliding speed by a factor ten, for example. Maybe the two main parts of this paper have been obtained independently and put together afterwards. Nevertheless, should the editor decide to publish this paper, I propose the following mandatory revisions:

Answer:

We strongly disagree with the referee's statement "friction results are in the borderline between EHL and mixed lubrication"! Even if one takes the definition of friction regimes literally (boundary lubrication: $\lambda \leq 1$, mixed lubrication $1 < \lambda \leq 3$, EHL: $3 < \lambda$) our values indicate a borderline between boundary and mixed lubrication. In our opinion, arguing about digits after the decimal point of λ is somewhat misleading. According to an excellent comment of Reviewer #4, we estimated the contribution of the hydrodynamic component μ_{HD} to the recorded friction coefficient μ_{exp} . The fact that the hydrodynamic friction is much smaller than the measured friction coefficient for all lubricants can only be explained by a frictional behaviour that is dominated by boundary lubrication. Therefore, also Reviewer #4 strongly supports our assessment that all our experiments are performed under boundary lubrication conditions (i.e., 3 out of 4 Reviewers agree with us). Meanwhile, we have presented our work at many conferences and received enthusiastic feedback by the "tribology community". Nobody ever questioned that we operate in the boundary lubrication regime.

We also strongly disagree with the Reviewer's suspicion that "two main parts of this paper have been obtained independently and put together afterwards". The QMD calculations were specifically custom tailored to understand the experimental observation.

2) Reviewer's comment:

Concerning glycerol case: The authors cite Matta's work in PRB 2008. However I found that there is another more detailed paper from Matta in 2008 (published in Lubrication Science). In this paper, the authors studied ta-C/ta-C tribopairs in glycerol at 80 °C at very low sliding speed (lambda ratio below 0.1 in pure BL conditions). The CoF is about 0.02-0.04. Also they ran tests in glycerol vapour phase at same temperature. The CoF is about 0.05 in well agreement with results in Fig. 8 of submitted paper which matches very well the

experimental conditions of the gas phase. So the results in this reference are in strong contradiction with the submitted paper with CoF 5 times higher. The authors must cite this “Lub. Science” paper and must explain why their results are in discrepancy with literature data.

Answer:

We already noticed the Lubrications Science paper by Matta et al. They performed reciprocating cylinder-on-flat sliding tests of ta-C lubricated with glycerol at 80 °C. However, the maximum Hertz contact pressure ($P_{\text{cmax}} = 0.8$ GPa) is three times larger than the contact pressure in our experiments. Moreover, their sliding tests were done in the gas phase, whereas our test was done in the liquid phase. Obviously, these differences in the experimental conditions make a big difference in the frictional behaviour. In the gas phase, less glycerol molecules come into the contact area and the larger contact pressure can lead to higher probabilities of cold welding. We think that it is dangerous to directly compare the two results. Instead, we found that experimental conditions ($P_{\text{cmax}} = 0.27$ GPa, $T = 80$ °C) and recorded friction coefficient ($\mu < 0.01$) in Matta’s PRB paper are quite comparable with those in our paper. We cited the Lubrication Science paper in the Introduction, and only mentioned that they observed ultralow friction under the higher contact pressure.

“Moreover, Matta et al.¹¹ observed ultralow friction in reciprocating cylinder-on-disc tests of ta-C boundary lubricated with pure glycerol. Interestingly, a reduction of contact pressure resulted in superlow friction¹².”

3) Reviewer’s comment:

Concerning oleic acid case: The authors cite Kano’s work with oleic acid. In this paper the Stribeck curve is shown and this is the proper way to define accurately the lubrication regime. It is clearly shown in Kano’s work that ta-C/ta-C in glycerol does not display superlubricity under boundary lubrication (at very low speeds). This is again in strong contradiction with the submitted paper where superlubricity is said to happen under BL. The authors must comment and explain why they did not define the Stribeck curve to justify the lubrication regime.

Answer:

In their Tribology Letters paper, Kano et al. used a rotating cylinder-on-flat tester, where the sliding speed is constant. This experimental setup is suitable to explore different friction regimes by varying the sliding speed. In contrast, we employed a pin-on-disc reciprocating tribometer. The sliding speed varies in the stroke and at the end of stroke the sliding speed goes to zero. The setups are completely different, and it is really hard to put them into the same context. We thus cited Kano’s Tribology Letter paper in the Introduction without going into further details.

Reviewer #4:

1) Reviewer’s comment:

Some simple calculations allow one to demonstrate that the lubricated contacts in this paper are indeed operating in the boundary regime. Using Hertzian theory for point contacts, one

can calculate that the maximum contact pressure is 0.23 GPa, the contact radius is 0.00032 m and the contact area is 0.32E-6 m². From the viscosity and pressure-viscosity coefficients given in Table S2, the viscosities at 0.23 GPa pressure can be determined to be: octadecane 15 mPa-s, stearic acid 38 mPa-s, oleic acid 26 mPa-s and glycerol 110 mPa-s.

With these relatively small viscosity increases due to pressure, the traction (friction) coefficient, if the contact were in a high lambda value hydrodynamic regime, can be found from $\mu = (\text{viscosity} \times \text{area} \times \text{velocity}) / (\text{film thickness} \times \text{load})$. For the four oils above, μ (hydrodynamic) is 0.0023, 0.0020, 0.0021 and 0.0029, respectively. The values the authors measured in their pin on disk rig were 0.0700 (short period of steady state before what looks like scuffing), 0.0620, 0.0080 and 0.0080 (extra zeros are added for easier visual comparison with the calculated results).

It is possible to subtract the hydrodynamic friction to determine the boundary friction contribution. This indicates that all four lubricants had a significant friction contribution from asperity interactions, and therefore operated in the boundary lubrication regime. For octadecane and stearic acid, the boundary friction contribution was 30 times that of the hydrodynamic contribution. For oleic acid and glycerol it was only 2-3 times, suggesting that these are good boundary friction reducers.

Answer:

We are grateful to the reviewer for this excellent suggestion. We reproduced the reviewer's estimations using $\tau = \eta\dot{\gamma}$, where τ is shear stress in the lubricant, η the viscosity and $\dot{\gamma}$ the shear rate. By estimating the shear rate using $\dot{\gamma} = \frac{v}{h_{\text{centre}}}$, we obtained the friction coefficients $\mu_{\text{HD}} (= \frac{\tau}{P} = \frac{\eta v A_{\text{Hertz}}}{w h_{\text{centre}}})$ of 0.0013, 0.0013, 0.0011 and 0.0015 for octadecane, stearic acid, oleic acid and glycerol, respectively. Importantly, these values are much smaller than the experimentally recorded friction coefficients for all lubricants (the $\mu_{\text{HD}}/\mu_{\text{exp}}$ values are presented in Supplementary Table 2). Since μ_{exp} includes the contribution of the hydrodynamic component and asperity interactions, in our systems the asperity interactions are dominant. We added the following paragraph to Supplementary Note 1.

“Since some of our calculated λ parameters were so close to unity (and for glycerol with $f = 10$ Hz even above unity), we report here an alternative approach to determine the friction regime. Under hydrodynamic lubrication conditions the shear stress τ in the lubricant is given by $\tau = \eta\dot{\gamma}$, where η is the effective viscosity in the Hertz contact zone and $\dot{\gamma}$ the shear rate. If we estimate the shear rate using $\dot{\gamma} = \frac{v}{h_{\text{centre}}}$, the friction coefficient μ_{HD} under hydrodynamic lubrication is given by $\mu_{\text{HD}} = \frac{\tau A_c}{w} = \frac{\eta v A_c}{w h_{\text{centre}}}$. The viscosities η of the liquids at the pressure P are calculated by $\eta = \eta_0 e^{\alpha P}$. Here, the maximum Hertz contact pressure $P_{\text{cmax}} = \frac{3w}{2\pi r_c^2} = 242$ MPa is used. For all lubricants, the estimated μ_{HD} values are much smaller (0.0011–0.0015) than our experimentally recorded friction coefficients (see Supplementary Table 2). Even for oleic acid and glycerol, μ_{HD} is less than one-fifth of μ_{exp} . Since the recorded frictional forces include the contribution from the hydrodynamic component and direct asperity interactions, it is clear that boundary lubricated asperity contact friction is the dominant contribution to the

frictional behaviour of ta-C observed in Fig. 1. Especially for glycerol it becomes apparent that more than 80% of the measured friction originates from boundary lubrication.”

2) Reviewer’s comment:

Note that pure hydrodynamic films at $\mu \sim 0.002$ are already in the range of super-lubricity and that the crankshaft, connecting rod and camshaft bearings and much of the piston-ring cylinder stroke is already operating in this very low friction regime with full hydrodynamic films. To argue in the introduction to the paper that reducing μ to <0.01 would improve fuel economy by 61% is a great exaggeration and hurts the credibility of the paper. The Erdemir and Holmberg paper is widely misused and misquoted.

Answer:

Thank you for sharing your opinion on the Erdemir and Holmberg paper. We also have our doubts about the 61% fuel saving. We removed this number in the revised manuscript.

3) Reviewer’s comment:

With regard to reviewer 1's comments about which film thickness to use, I agree that it should be the central value not the minimum value when discussing friction, since that is where the majority of the load is carried. However, determining a lambda value is tricky. The roughness value is a function of the length of surface measured. Sayles** found that over many length scales the rms roughness increased with the square root of the length measured. An appropriate length in tribology is the width of the contact, which in this case is 640 um (see paragraph 1 above). The authors measured a 50 um length with the AFM so they should multiply their rms values by $(640/50)^{0.5}$ or 3.6. Doing this would lower all their lambda values by 3.6x and so they are all clearly in the boundary regime, regardless of whether minimum or central film thickness is used.

Answer:

We asked a couple of tribologists and obtained different opinions about the correct film thickness in the formula for the λ parameter. There were the ‘minimalists’ that prefer the minimum film thickness h_{min} , since friction is dominated by the zones with highest friction coefficient and there were ‘centralists’ that opt for h_{centre} , since that is where the majority of the load is carried. In this final revision, we follow Reviewers #1 and #4 and join the ‘centralists’ simply because the λ values obtained from h_{centre} are the more conservative estimates. Thus, we removed the results of the minimum film thicknesses, and instead used the central film thicknesses for λ values.

We appreciate Reviewer #4’s suggestion to use larger roughness values in the calculations of the λ parameter by upscaling the roughness measured on a small area to the roughness in the much larger Hertz contact area. Unfortunately, this does not work for our surface topographies. Reviewer #4 quotes the work of Sayles and Thomas from late 1970s. This pioneering work is one of foundations for today’s scaling theory of self-affine surfaces. This is based on the power spectral density (PSD) of the surface modes. The PSD $C(q)$ scales as q^{-2-2H} over many decades in the wave vector $q (= \frac{2\pi}{\lambda})$. Here, H is the Hurst exponent. However, usually this scaling has a lower limit q_l and $C(q)$ rolls off (i.e. becomes roughly

constant) for $q < q_l$. If the size L of the scanned area is small enough, so that $q_L = \frac{\pi}{L} > q_l$ the root-mean-square (RMS) roughness scales as $R_{RMS} \sim L^H$. In this case the Reviewer's suggestion would be correct. However, we checked the PSD of ta-C surfaces after sliding with glycerol and realized $q_L \ll q_l$ for the relevant L values (i.e. $L \approx 50 \mu m$ and $L \approx 600 \mu m$). Under these conditions the Hurst scaling breaks down and roughness is almost independent of the scan length. Therefore, in the revised version of the manuscript we still use our measured roughness without scaling. The following explanation about the treatment of the RMS roughness scanned over a small area was added to Supplementary Note 1.

“In principle, the roughness values R_{RMS} entering Eq. (8) should be measured by scanning a surface with the size of the contact area $A_c = \pi r_c^2$ between pin and disc. In our experiments, the contact radius $r_c = R_{pin} \left(\frac{3\pi P_N}{4E'} \right)^{\frac{1}{3}}$ is given by $r_c = 314 \mu m$ and therefore the size of A_c makes such a roughness measurement difficult (especially on the pin). Thus, the roughness has to be estimated from scans over smaller areas of the size A_{mic} . The validity of these estimates crucially depends on the power spectral density (PSD) of the surfaces⁵. The PSD of most technical surfaces scales as $C(q) = C(q_l) \left(\frac{q}{q_l} \right)^{-2-2H}$ for $q_l < q < q_s$. Here, H represents the Hurst exponent that usually ranges between 0.7 and 0.9⁵, q is the wave vector and q_s is the small and q_l is the large wavelength cutoff with $q_l \ll q_s$. The RMS roughness is determined by the PSD via $R_{RMS}^2 = \frac{1}{2\pi} \int_{q_L} q C(q) dq$, where $q_L = \frac{\pi}{L}$ with L the length of the scanned surface area. For $q_L = q_c = \frac{\pi}{L_c} > q_l$ the simple scaling law $R_{RMS,c} = R_{RMS,mic} \left(\frac{L_c}{L_{mic}} \right)^H$ holds (with $L_c = \sqrt{A_c}$ and $L_{mic} = \sqrt{A_{mic}}$) indicating that $R_{RMS,c} > R_{RMS,mic}$. Consequently, using $R_{RMS,mic}$ in Eq. (8) would yield overestimated λ values. For the surfaces considered in this article $H \approx 0.85$, $q_l \approx 1.6 \mu m^{-1}$ and $C(q_l) = 0.004 \mu m^4$. Therefore, $q_c = 0.0056 \mu m^{-1}$ and $q_{mic} = \frac{\pi}{L_{mic}} = 0.043 \mu m^{-1}$ are much smaller than q_l , i.e. $q_c \ll q_{mic} \ll q_l$. In this case, since $C(q)$ is about constant for $q < q_l$, the RMS roughness is given by

$$R_{RMS}^2 = \frac{C(q_l)}{4\pi} \left[\left(1 + \frac{1}{H} \right) q_l^2 - q_L^2 \right]. \quad (9)$$

Inserting $q_L = q_{mic}$ in Eq. (9) results in $R_{RMS,mic} = 42.106 nm$, while $q_L = q_c$ reveals $R_{RMS,c} = 42.113 nm$. Thus, $R_{RMS,c} \approx R_{RMS,mic}$ and therefore using $R_{RMS,mic}$ in Eq. (8) is the valid choice for the surfaces considered in our experiments.”

For all C_{18} lubricants, we obtained λ values below one using h_{centre} (see Supplementary Table 2). The λ value for glycerol is only slightly larger than unity ($\lambda = 1.30$) indicating a borderline situation between boundary and mix lubrication. However, as we already showed in the previous revision, the friction coefficients at lower frequencies (at $f = 4$ and 5 Hz,

ensuring $\lambda < 1$) are the same as that at the original frequency ($f = 10$ Hz). While the contribution of asperity interactions to the experimentally recorded friction coefficients increases with decreasing f (Supplementary Table 2), the friction coefficient stays constant at the superlubricity level. This means that all experiments at $f = 4$ – 10 Hz were dominated by asperity interactions and probing the same (boundary) lubrication regime. This assessment is corroborated by the calculation of the hydrodynamic contribution to friction suggested by Reviewer #4 that shows that less than 20% of the friction results from hydrodynamic losses and therefore we completely agree with Reviewer #4 that all our experiments are clearly in the boundary lubrication regime.

4) Reviewer's comment:

The authors could have checked the hydrodynamic contribution by running steel on steel pin on disc experiments with superfinished surfaces (5 nm rms) to confirm that their "rough" ta-C coatings are running in the boundary regime. However, I don't think that is necessary for publication.

Answer:

Thanks for the suggestion. We will set up the system in the near future since we are interested in the transferability of our findings to other materials.

5) Reviewer's comment:

In table S2, "curvature" should be "radius of curvature" as curvature is $1/\text{radius}$ and has units of m^{-1} .

Answer:

We corrected it.